# Molecular Mechanisms of Drug Resistance in Clear Cell Renal Cell Carcinoma

**DOI:** 10.3390/cancers17101613

**Published:** 2025-05-10

**Authors:** Nicoletta Bianchi, Pietro Ancona, Gianluca Aguiari

**Affiliations:** 1Department of Translational Medicine, University of Ferrara, 44121 Ferrara, Italy; nicoletta.bianchi@unife.it (N.B.); pietro.ancona@unife.it (P.A.); 2Department of Neuroscience and Rehabilitation, University of Ferrara, 44121 Ferrara, Italy

**Keywords:** chemotherapy resistance, HIF, p53, non-coding RNA, autophagy, ferroptosis

## Abstract

Metastatic renal carcinoma (mRCC) is very hard to cure, despite the use of new-generation therapies that include inhibitors of key proteins such as tyrosine kinases (TK) and blockers that stimulate the immune response against cancer cells called immune checkpoint inhibitors (ICIs). The efficacy of chemotherapy decreases over time because of innate and acquired resistance that dramatically reduces the life expectancy of mRCC patients. This review aims to analyze the molecular mechanisms involved in cancer drug resistance in order to identify new therapeutic targets that could overcome this matter. We discussed the main pathways involved in pharmacological resistance including angiogenic factors, oncogenes, tumor suppressor proteins, vesicular transport, ferroptosis, and non-coding RNAs. Clinical trial studies targeting factors involved in resistance or multi-target therapies are also reported. The results discussed could suggest the setting of specific treatments with improvements for subjects suffering from mRCC and could help clinicians choose the best options for the management of ccRCC patients.

## 1. Introduction

Renal cell carcinoma (RCC) is the most common urologic cancer and accounts for about 3% of all human tumors [1]. Clear cell renal cell carcinoma (ccRCC) is the most frequent histotype and is associated with the greatest disease severity, developing metastases in one-third of patients [1,2]. Metastatic kidney carcinoma (mRCC) is hard to cure despite the advancements made in the therapeutic field by using multi-target approaches. Currently, first-line therapy for mRCC is performed by the treatment with a combination of immune checkpoint inhibitors (ICIs) alone or combined with tyrosine kinase inhibitors (TKIs) [3]. In particular, mRCC patients may be treated with a combination of nivolumab and ipilimumab which are antibodies against PD-1 and CTLA-4 receptors, respectively [3]. Alternatively, ICIs can be used in combination with TKI, for instance, pembrolizumab (PD-1 inhibitor) and axitinib (anti-VEGF); nivolumab (anti-PD-1) and cabozantinib (TKI); pembrolizumab (anti-PD-1) and lenvatinib (inhibitor of VEGFR receptors and other RTKs); avelumab (anti-PD-L1) and axitinib (anti-VEGF) [3,4]. First-line monotherapy with anti-angiogenic TKIs may be used for mRCC patients who are intolerant to ICI treatment [4]. For patients who show disease progression after ICI, further treatment is needed following first-line ICI-TKI-based therapy. Second-line therapy for these subjects involves different treatments, including ablative radiotherapy, metastasectomy, and pharmacological treatments [4]. Second-line treatment may be achieved by TKI monotherapy or TKI in combination with mTOR inhibitors such as everolimus [4]. However, the overall survival (OS) is still limited and disease progression in several patients is frequently observed [5]. Resistance to ICIs- and TKIs-based therapies, either intrinsic or acquired, remains a significant challenge [6,7].

In this review, we present the latest findings on the molecular mechanisms of drug resistance employed by cancer cells to evade chemotherapy-induced cell death, as there has been limited discussion on this topic. Cancer cells use different ways to elude cell death during prolonged treatment with anticancer therapies. Drug resistance mechanisms include the enhancement of anti-vascular and survival pathways, cell adaptation to the harmful microenvironment, and epithelial-to-mesenchymal transition (EMT). We have thoroughly explored these aspects with particular emphasis on signaling pathways, biological processes, and non-coding RNA (ncRNAs) expression involved in drug insensitivity. NEXT-generation drugs and multitarget therapies designed to bypass resistance mechanisms, along with clinical trial results, are also discussed. Mechanisms of resistance related to inflammation and the immune system have been extensively covered in the literature and, therefore, are not included in this work.

## 2. Materials and Methods

Rigorous bibliographic research in the field of pharmacological resistance in kidney cancer has been performed using the following query: (“renal cell carcinoma” [All Fields] AND “drug resistance” [All Fields] NOT (Review [pt]) NOT (Meta-analysis [pt]) NOT (“Systematic Review” [pt]). We initially obtained 835 manuscripts, which were then reduced to 643 by selecting articles published between 2015 and 2025, the period during which the Gaussian distribution reached its highest mean. By excluding those not specifically focused solely on renal carcinoma, we selected 583 manuscripts, listed along with the sources (PMID, Title, Authors, Citation, First Author, Journal/Book, Publication Year, Create Date, PMCID, NIHMS ID, DOI) in Appendix A. We checked all articles by manual curation to verify relevance and adherence to the topic, excluding those not in line. Some research and review articles were manually screened and inserted in the work drafting.

Concerning the molecular mechanisms in which ncRNAs are involved, we found 50 papers discussing the role of microRNAs, 15 dealing with long non-coding RNAs (lncRNA), and 4 engaging circular RNAs (circRNAs). Since the mechanisms involving microRNAs have been extensively covered in the literature, this paragraph focuses solely on the remaining mechanisms.

Overall, we selected 130 papers that align best with the scope of the special issue and highlight the most relevant findings in the field of pharmacological resistance in kidney cancer research.

## 3. Results and Discussion

Drug resistance can be defined as intrinsic or innate (primary) and acquired (secondary) [7]. Intrinsic resistance is mainly due to the original insensitivity of the tumor to pharmacological treatment. Conversely, acquired drug resistance occurs after an initial response to therapy, but later tumor cells become insensitive to treatment because of several resistance mechanisms that promote cancer progression. Intrinsic resistance may be due to multiple factors including tumor cells unresponsive to therapy, off-target chemotherapy, and drug inactivation [8]. Chemotherapy may induce acquired resistance when the treatment leads to the acquisition of new mutations or the selection of pre-existing drug-resistant clones [8]. This problem could be overcome by administering drugs that act on different targets.

In this review, we pay particular attention to molecular mechanisms such as hypoxia-inducible factor (HIF) activation, p53 alteration, autophagy modulation, drug removal, ferroptosis suppression, and lncRNA dysregulation, which may induce therapy resistance in advanced kidney cancer.

### 3.1. HIF Activation

Constitutive activation of HIF is observed in the majority of ccRCC patients due to suppression of the von Hippel–Lindau (VHL) protein [9]. HIF activation contributes to cancer vascularization and progression making it druggable to improve cancer treatment. Despite different HIF inhibitors being designed and tested in advanced renal cancer, poor improvements in progression-free survival (PFS) and OS were obtained [9]. In this regard, treatments with HIF2α inhibitors including FDA-approved belzutifan in pVHL-mutated patients with ccRCC were associated with an overall response rate of 49% and a clinical benefit rate of 98%. After 24 months of treatment, the percentage of progression-free patients was 96% [10]. Nevertheless, both intrinsic and acquired resistance to HIF-2α inhibitors has been observed in ccRCC, particularly with the use of first-generation inhibitors such as PT2385 and PT2399 [10]. Moreover, selected pre-existing resistant clones after pharmacological treatment, as a mechanism of intrinsic resistance, could explain the rapid disease progression observed in some treated patients [10]. The treatment with the first-class HIF2α inhibitor PT2385 that promotes the dissociation of the HIF2α/HIF1β complex leads to the inhibition of HIF2α-dependent gene expression resulting in angiogenic process blockage [11]. However, prolonged PT2385 treatment results in acquired resistance due to the generation of a gatekeeper mutation (G323E) in HIF2α, which prevents drug binding making the therapy ineffective [11]. Alternatively, this mutation could be a somatic lesion carried by a few cells into the tumor and thus undetectable before the treatment. As mentioned above, this would be another intrinsic resistance mechanism that emerged by drug treatment which leads to the selection and amplification of resistant clones that promote tumor progression. A similar mechanism of action is exhibited by the HIF2α inhibitor PT2399, which shows greater activity than sunitinib and is better tolerated. Even then, the treatment with this compound has caused pharmacological resistance in some ccRCC [12]. These effects are due to mutations in the binding site of HIF2α and HIF1α that preserve HIF2α/HIF1β interaction despite the presence of the antagonist PT2399, deleting the inhibitory effect of this compound [12]. Interestingly, resistance to therapy may occur despite the drug preventing HIF2α interaction with HIF1β resulting in HIF2α inhibition and circulating erythropoietin suppression, suggesting that tumor progression is HIF2α independent. These observations indicate that other players may intervene in mechanisms associated with the development and progression of renal cancer and other studies are needed to elucidate the nature of this tumor.

Long-term treatment with HIF2α inhibitors causes not only drug resistance but also some side effects including anemia in 90% of patients mainly due to the reduction in erythropoietin production and fatigue in 64% [10]. New inhibitors to overcome drug resistance problems should be designed and tested or a combination of more agents could improve current therapies in ccRCC patients. Some HIF2α inhibitors, such as DFF332, require further investigation to establish a recommended dosing regimen, evaluate their efficacy and safety, and explore their full potential as combination therapy partners [13].

By inhibiting HIF, it is possible to disrupt processes depending on hypoxic conditions, especially in combination with immune checkpoint inhibitors (ICIs) or tyrosine kinase inhibitors (TKIs). This represents a promising therapeutic strategy of ICIs by reversing the immunosuppressive milieu and improving the response to TKIs by limiting VEGF-mediated angiogenesis. Preclinical studies and early-phase clinical trials have started to explore these combinations, showing synergistic effects and supporting further investigation in ccRCC characterized by hypoxia-driven resistance mechanisms.

Currently, a first-line setting (phase 2 LITESPARK-003 study) by using a double treatment with the HIF2α-VEGF inhibitor (belzutifan) in combination with the TKI cabozantinib has shown manageable toxicity [14]. These findings provide the rationale for further randomized trials using belzutifan in combination with TKIs to evaluate improvements in PFS and OS in ccRCC patients.

Phase III clinical trials are currently being conducted to compare the efficacy of belzutifan versus everolimus in patients with advanced RCC previously treated with anti-PD-(L)1 antibodies and VEGF TKIs. These studies demonstrate that belzutifan ameliorates the global health status and quality of life of patients and shows superior efficacy compared to everolimus [15]. Other trials are ongoing: (i) LITESPARK-024 to evaluate the efficacy of belzutifan, also in combined therapy with palbociclib [16]; (ii) LITESPARK-011, investigating the effects of belzutifan combined with lenvatinib versus belzutifan combined with cabozantinib in patients who did not respond to anti-PD-1/PD-L1 therapy [17]; (iii) belzutifan administrated with pembrolizumab in patients who have undergone nephrectomy and/or metastasectomy [18].

Other inhibitors may also be of interest, such as modulators of HIF factors under hypoxic conditions. FTY720, an inhibitor of the S1P signaling pathway that triggers HIF1α and HIF2α expression, reduces their levels in several cancer cells as well as in different mouse models for ccRCC [19].

Since the activation of HIF2α may itself cause resistance to the conventional therapy, an alternative strategy may involve targeting genes that are modulated by this transcriptional factor. Indeed, it was observed that this transcription factor is able to promote the expression of polo-like kinase 1 (Plk1) leading to sunitinib resistance in vitro, in vivo, and in patients with metastatic ccRCC [20]. Another mechanism of drug resistance involving HIF2α in kidney cancer is due to the deacetylation of the protein codified by ETS proto-oncogene 1 (ETS1) through the activation of the histone deacetylase 8 (HDAC8) [21]. HDAC8 deacetylates ETS1 at the K245 site fostering the interaction between this transcription factor and HIF2α. The activation of the ETS1/HIF2α complex results in a decreased sensitivity to treatment with sunitinib in ccRCC cells. Interestingly, the application of TKI in ccRCC cells causes the increased expression of HDAC8 by preventing the phosphorylation of signal transducer and activator of transcription (STAT3), suggesting thus a mechanism of acquired resistance adopted by cancer cells following the treatment with sunitinib [21]. The transcriptional activity of HIF2α is also potentiated by the interaction with the chromatin helicase-DNA binding protein 1L (CHD1L), which is overexpressed in ccRCC and correlates with poor prognosis [22]. CHD1L interacts with HIF2α and causes sunitinib resistance via sustaining VEGFA. The inhibition of CHD1L removes sunitinib resistance and strengthens the anticancer effect of this drug [22].

These observations provide new opportunities to test possible therapeutic targets including Plk1, HDAC8, and CHD1L that could enhance the effectiveness of treatments in mRCC patients. HIF-mediated pathways that are strongly involved in ccRCC are currently targeted and many studies were conducted not only in models for mRCC but also in clinical trials. As reported above, several trials are now running and findings obtained at the end of these studies will indicate if these strategies will yield further improvements for mRCC patients. Last but not least, we will evaluate the side effects that could compromise therapy response, which will require careful management for future clinical applications.

### 3.2. p53 Dysfunction

It is well known that p53 loss of function (LOF) is involved in tumor progression in all cancers. In kidney carcinoma, p53 mutations are rarely detected; however, various mechanisms contribute to the degradation of the p53 protein, promoting disease progression and therapy resistance. p53 may be inactivated by the enhanced activity of HIF-2α, resulting from VHL loss of function (LOF), which leads to Hdm2-mediated suppression of p53 [23]. The inhibition of Hdm2 or the downregulation of HIF2α restores p53 function and reverses the resistance of ccRCC cells to chemotherapy-induced cell death [23]. Consistently, other studies report that the inhibition of MDM2, a negative regulator of p53, enhances the anticancer activity of rapamycin in ccRCC cells. MDM2 is upregulated in rapamycin-resistant cells which exhibit low levels of p53 protein. Inhibition of MDM2 re-sensitizes ccRCC cells to rapamycin through activation of p53 [24]. In addition, the combined treatment with the TKI sorafenib and the MDM2 inhibitor nutlin-3 shows a synergistic effect promoting cell death by activating different pro-apoptotic proteins, including p53, PUMA, and Bax. Conversely, this treatment inhibits the vascular endothelial growth factor receptor-2 (VEGFR-2), ERK kinases, and the anti-apoptotic protein Bcl-2 in ccRCC cells [25]. Interestingly, another study shows that combined treatment with sunitinib and nutlin-3 suppresses cell proliferation by restoring p53 function [26]. The inactivation of p53 causes reduced sensitivity to sunitinib and, as observed in a phase III trial (Javelin 101 study), p53-mutated ccRCC patients exhibit a lower PFS after first-line treatment with sunitinib, suggesting that p53 is involved in chemotherapy resistance also in kidney cancer [26]. Furthermore, the expression of VHL in ccRCC cells re-sensitizes these cells to adriamycin (ADM) and sunitinib treatment, while the knockdown of VHL induces chemoresistance against these drugs. Importantly, the combined expression of VHL and p53 inhibits cell proliferation and promotes apoptosis following treatment with ADM or sunitinib, likely through downregulation of HIF expression [27]. Another mechanism of p53 inactivation and drug resistance involves the overexpression of the promyelocytic leukemia (PML) protein in ccRCC tissues. PML acts as a negative regulator of p53 by a not yet well-clarified molecular mechanism; nevertheless, the depletion of PML leads to the accumulation of p53 protein and its downregulation promotes p53-dependent cellular senescence [28]. Notably, treatment with the FDA-approved PML inhibitor arsenic trioxide induces PML degradation, thereby increasing p53 levels and inhibiting ccRCC expansion both in vitro and in vivo [28].

These data indicate that HIF plays a central role in ccRCC carcinogenesis and drug resistance by a mechanism that involves the inhibition of p53 function. In cancer, p53 mutations are associated with poor prognosis and therapy resistance. Stably expressed p53 mutants not only lose the tumor-suppressing function of the wild-type form but also acquire tumor-promoting activities, including increased cell proliferation, migration, and drug resistance [29]. Most drugs induce cell death via p53 activation, but in the case of p53 LOF or gain of function (GOF), its activity is lost, resulting in ineffective therapy.

Based on these data, approaches addressed to restore p53 expression should be carefully evaluated in order to prevent the expression of p53 mutant, which is associated with poor outcomes. The identification of p53 mutations might lead to the generation of alternative therapeutic strategies able to repair p53 lesions or activate the immune system against p53 mutants, killing cancer cells and inducing cancer regression. In ccRCC, the expression of this tumor suppressor is downregulated; therefore, the reactivation of wild-type p53 could enhance the therapeutic efficacy in patients refractory to pharmacological treatments. Several clinical trials are being performed or are ongoing to achieve this goal. A phase I study (PYNNACLE) [30] is focused on the PC14586 (rezatapopt) used to correct specifically the mutation Y220C of the TP53 for restoring its normal conformation [31,32]. The efficacy of this compound was checked in patients affected by a variety of solid tumors, showing a decreased number of circulating tumor cells after treatment. Further investigations are conducted to test the effects of the combined treatment of PC14586 and pembrolizumab [33]. A joint US–China study is currently underway on the same mutation to evaluate the safety and efficacy of JAB-30355 [34]. Other strategies could be developed, based on the design and construction of TP53-Y220C neoantigens to enhance the affinity and binding stability to HLA-A0201 molecules to induce increased production of cytotoxic T lymphocytes (CTLs), indicating an improvement in immunogenicity [35,36].

### 3.3. Akt-mTOR Signaling

Beyond the abnormal activation of HIF transcription factors, signaling pathways driven by altered activation of mTOR are involved in ccRCC pathogenesis. Several therapies targeting mTOR have been approved by the FDA for the treatment of advanced ccRCC. However, the efficacy of temsirolimus and everolimus administration, both mTOR inhibitors (rapalogs), is limited due to the acquisition of resistance [37]. Prolonged treatment with rapalogs may induce adaptive resistance in mRCC patients by activating mTORC2/Akt signaling and pro-survival ERK and STAT3 pathways. In addition, enhanced hypoxic signals are also involved in the pathogenesis and drug resistance of ccRCC [38]. Similarly, another study reports that acquired resistance to temsirolimus in human renal carcinoma cells occurs through the activation of mTORC2. The triggering of signal transduction pathways via mTORC2, but not via mTORC1, leads to the Akt and p44/42 mitogen-activated protein kinase (MAPK) phosphorylation following temsirolimus treatment in resistant ACHN kidney cancer cells. Therefore, these signaling pathways seem to play a role in the development of a resistant phenotype to rapalogs in ccRCC cells [39]. Conversely, Yang J. and colleagues report that mTORC1, abnormally activated in ccRCC, promotes cell growth and cancer progression. As previously observed, long-term treatment with mTOR inhibitors desensitizes cancer cells to drugs, often by mutation of target proteins. However, mTOR mutations are rarely found in ccRCC tissues, and other resistance mechanisms must be involved [37]. Resistance to prolonged rapalog treatment in tumor cells, despite persistent mTORC1 inhibition, has been observed, while mTORC1 is reactivated in non-tumor cells that constitute the tumor microenvironment (TME). Thus, the activation of mTORC1 observed in TME is the likely cause of resistance to rapalogs, and these findings highlight the importance of the TME in mediating antitumor drug efficacy [37]. Activation of mTOR also plays an important role in resistance to TKI treatment. It has been reported that ccRCC cells resistant to sunitinib exhibit elevated levels of the palmitoyl acyltransferase ZDHHC2 that mediates AGK S-palmitoylation and promotes the translocation of Acylglycerol kinase (AGK) into the plasma membrane. Membrane AGK induces the activation of the PI3K–Akt–mTOR signaling pathway, contributing to sunitinib resistance in cells and mouse models for ccRCC [40].

Because treatment with rapalogs has shown resistance after prolonged use, research and development of more efficient mTOR inhibitors are needed. In this regard, the treatment with Rapalink-1, a drug that links rapamycin and the mTOR kinase inhibitor MLN0128, has been tested with optimal anticancer effects in breast cancer cells carrying mTOR resistance mutations [41]. Accordingly, in both in vitro and in vivo ccRCC models, treatment with Rapalink-1 shows greater efficacy than temsirolimus and leads to deactivation of the mTOR pathway, including p70S6K, 4EBP1, and Akt. Moreover, the application of Rapalink-1 in sunitinib-resistant 786-O ccRCC cells inhibits not only the mTOR signaling but also the MAPK and ErbB pathways as well as ABC transporters, which are known to confer resistance to several drugs [41].

More recently, the combination of the mTORC1/2 inhibitor sapanisertib and the TKI cabozantinib have effectively inhibited tumor growth in patient-derived xenografts (PDXs), some of which were resistant to conventional TKI and immunotherapy combinations. Their action appears to modulate the ERK pathway and downstream transcription factors, leading to cell cycle arrest and apoptosis [42]. Although it appears to be less active when administered alone in refractory mRCC [43] and did not improve the effects of everolimus in advanced or refractory ccRCC [44], sapanisertib has been shown to be effective in the treatment of solid tumors, including ccRCC, in a phase I clinical study when combined with metformin [45]. Sapanisertib acts by inhibiting the mTOR/AKT/PI3K pathway and shows efficacy in combination with carboplatin and paclitaxel [46] when different pathways are concurrently targeted.

Notably, autophagy is one of the processes regulated by mTOR; in particular, the activation of this kinase inhibits autophagy and promotes growth signals. However, anticancer drugs that inhibit mTOR cause the induction of autophagy, which may contribute to cancer progression helping cancer cells to recover energy and remove chemotherapy agents, as widely described below. Therefore, the combined treatment with mTOR and autophagy inhibitors could overcome this issue, limiting the pharmacological resistance.

### 3.4. MEK–ERK Pathway

Mechanisms of TKI resistance also involve MEK/ERK signaling; MAP2K1 and MAP2K2 (known as MEK1/2) are central components of this pathway. These kinases connect signals from different upstream regulators including SOS, RAS, and Raf, and downstream effectors such as ERK1/2, functioning as “gatekeepers” of ERK1/2 [47]. MAP2K2 upregulation in ccRCC tissues induces resistance to sunitinib, axitinib, and sorafenib, three well-known inhibitors of VEGFR, in a mechanism involving the overexpression of SP1 and leading to the activation of the MEK/ERK pathway. The combination of MEK and VEGFR inhibitors markedly enhances the sensitivity of ccRCC cells resistant to TKI inducing a strong cell death [47]. Another study has demonstrated that the interferon-induced transmembrane protein 3 (IFITM3) is overexpressed in ccRCC samples after TKI administration and correlates with TKI resistance [48]. IFITM3-related resistance to TKI treatment is associated with the activation of TNF receptor-associated factor 6 (TRAF6) and MAPK/AP-1 pathways. The increased expression of IFITM3 in TKI-resistant ccRCC cells causes an aggressive phenotype inducing increased proliferation, survival, migration, and invasion. Interestingly the TKI-resistant phenotype is reversed by suppression of IFITM3 expression, making it a draggable element [48]. In addition, another report demonstrates that the transcription gene Sry-related high-mobility group (HMG) box 9 (SOX9) can activate the Raf/MEK/ERK pathway resulting in sorafenib/sunitinib resistance in ccRCC cells and tissues [49]. SOX9 downregulation re-sensitizes ccRCC cells to TKI treatment by deactivating Raf/MEK/ERK signaling. The treatment with sunitinib in mRCC patients negative for SOX9 expression shows a better response compared with those expressing SOX9, confirming that the activation of MEK/ERK signaling pathways is involved in drug resistance [49]. Consistently, the increased activity of MEK/MAPK signaling after sunitinib treatment induces TKI resistance in ccRCC tumors activating cellular invasion, inflammatory response, and immune cell trafficking genes as well as the accumulation of myeloid-derived suppressor cells (MDSC). The combined treatment with the MEK inhibitor PD-0325901 and sunitinib inhibits phospho-MEK1/2, phospho-ERK1/2, and MDSC and abrogates TKI resistance increasing the anti-cancer efficacy of these drugs [50]. Moreover, it was described that the membrane-associated guanylate kinase, WW, and PDZ domain-containing protein 3 LOF codified by the MAGI3 gene induce the activation of ERK signaling contributing to cancer progression and sunitinib resistance in ccRCC patients. MAGI3 interacting with the MAS receptor cooperates with the Ang-(1-7)/MAS/ERK axis regulating cell proliferation in ccRCC cells. Patients expressing low levels of MAGI3 exhibit poor prognosis and TKI resistance compared with those with higher MAGI3 expression, suggesting that MAGI3 might work as a tumor-suppressor gene improving sunitinib sensitivity in patients who express this enzyme [51]. Furthermore, glycolysis enzymes may also contribute to the progression of kidney cancer and drug resistance. It was reported that the upregulation of phosphoglycerate kinase 1 (PGK1) causes the increase in glycolysis-related enzymes and C-X-C chemokine receptor type 4 (CXCR4) that induce CXCR4-mediated phosphorylation of Akt and ERK leading to sorafenib resistance [52].

Taken together, these reports indicate that the MEK–ERK signaling pathway is strongly involved in cancer progression and therapy resistance; therefore, the targeting of these kinases and their associated signaling could improve current therapies for advanced kidney carcinoma. In this regard, a phase II open-label multicenter, multicohort study including ccRCC patients was carried out by using the MEK inhibitor cobimetinib in combination with atezolizumab. This treatment has shown weak activity in anti-PD-1/PD-L1 treatment-naive renal cell carcinoma and no activity in checkpoint inhibitor-treated patients [53]. These findings indicate that further trials with other MEK inhibitors alone or in combination should be assessed to improve therapeutic efficacy.

### 3.5. Wnt/β-Catenin Signaling

The Wnt family consists of 19 glycoproteins that modulate cell proliferation, differentiation, survival, migration, and stem cell self-renewal. Wnt signaling and its target genes are involved in various biological processes, including the development of renal cancer. In ccRCC, Wnt expression is associated with increased tumor size, advanced cancer stage, progression, migration, and chemoresistance [54]. In particular, Cai and colleagues demonstrated that in ccRCC cells resistant to sunitinib, increased activation of Wnt/β-catenin signaling is associated with tumor growth and metastasis [55]. In this regard, inhibition of β-catenin signaling by treating ccRCC cells with ovatodiolide decreases cell viability, survival, migration, and invasion, as well as tumor growth in xenograft mouse models [56]. Ovatodiolide reduces β-catenin phosphorylation, preventing its nuclear translocation, decreasing protein stability, and impairing its association with the transcription factor 4. Interestingly, the combined treatment with ovatodiolide and sorafenib or sunitinib overcomes drug resistance in TKI-resistant ccRCC cells [56]. Another study reports that the overexpression of Baculoviral IAP repeat-containing 6 (BIRC6) enhances proliferation, migration, and invasion of cultured ccRCC cells in a mechanism involving the upregulation of CXCR4 and β-catenin pathway activation [57]. BIRC6 overexpression is linked to increased sunitinib resistance in ccRCC cells by Wnt/β-catenin-mediated signals. Pharmacological treatment with the Wnt/β-catenin inhibitor XAV-939, as well as β-catenin silencing, reduces cell growth, tumor sphere formation, migration, and invasion in ccRCC cells. Moreover, the treatment with XAV-939 markedly inhibits tumorigenesis in a BIRC6-overexpressing xenograft RCC mouse model [57]. Accordingly, the activation of Wnt3a and Frizzled1 (FZD1)-related signaling contributes to drug resistance in kidney cancer. In particular, it was observed that the protein disulfide isomerase family 6 (PDIA6), elevated in renal carcinoma cells and tissues, induces imatinib resistance through a mechanism involving the Wnt3a–FZD1 axis. The silencing of PDIA6 reduces Wnt3a–FZD1 expression and re-sensitizes ccRCC-resistant cells to imatinib treatment [58].

These findings suggest that many pathways regulated by Wnt/β-catenin signaling contribute to drug resistance in ccRCC, and the pharmacological targeting of these elements could re-sensitize cancer cells to canonical treatment. Currently, no clinical trials are running using Wnt modulators in kidney cancer. Most of the data were obtained in in vitro and in vivo models limiting the translational medicine and the therapeutic potential of these tumor-associated targets.

### 3.6. Other Pathways

Mechanisms of drug resistance involve other signaling pathways including STAT, mesenchymal-epithelial transition factor (cMet), and platelet-derived growth factor receptor (PDGFR). STAT1 is overexpressed in ccRCC tissues and contributes to chemotherapy resistance. The suppression of STAT1 inhibits cell growth in both in vitro and in vivo models for ccRCC. Moreover, the inhibition of STAT1 re-sensitizes ccRCC cells to radiotherapy and Taxol treatment [59]. Furthermore, Lu D. and colleagues report that STAT2 promotes the expression of the solute carrier family 27 member 3 (SLC27A3), involved in pazopanib resistance in ccRCC cells. SLC27A3 knockdown overcomes pazopanib insensitivity; therefore, SLC27A3 and STAT2 could represent new therapeutic targets for mRCC treatment [60]. Also, activation of the STAT3 pathway contributes to sunitinib resistance in ccRCC; in fact, it was reported that the upregulation of the circular RNA circPTPN12 in ccRCC tissues was associated with poor prognosis. Mechanistically, the overexpression of circPTPN12 induces the activation of the STAT3 signaling pathway enhancing proliferation, migration, invasion, and sunitinib resistance [61]. In ccRCC tissues, STAT3 may be activated by the overexpression of gankyrin leading to the upregulation of the C-C motif chemokine ligand 24 (CCL24) that generates a positive loop enhancing the expression of gankyrin and STAT3 via C-C chemokine receptor type 3 (CCR3). Increased levels of gankyrin stimulate proliferation, invasion, migration, and pazopanib resistance. Gankyrin knockdown or CCR3 inhibition reverses the resistance to pazopanib and prevents metastasis in in vivo models for ccRCC [62].

Another mechanism of TKI resistance involves the upregulation of the oncogene cMet. As already reported, TKIs were widely used for systemic therapy in mRCC patients; however, pharmacological resistance phenomena following sunitinib and sorafenib treatment were observed. In particular, resistant ccRCC cells exhibit the activation of the cMet receptor IRAK1 and the reduction in the tumor suppressor MCPIP1 expression, triggering the metastatic process [63]. Moreover, the prolonged treatment with sunitinib enhances the expression of both MET and AXL metastasis-related receptors in ccRCC cell lines increasing cell migration and invasion. The inhibition of AXL and MET treating cells with the multi-kinase inhibitor cabozantinib overcomes resistance due to long-term treatment with sunitinib in ccRCC cells [64]. Another study reports that the prolonged treatment with sunitinib in a patient-derived ccRCC xenograft model causes the increased expression of cMet factor, which is involved in acquired drug resistance. The treatment with axitinib, a well-tolerated tyrosine inhibitor, reduces tumor growth in sunitinib-resistant ccRCC tissues. Moreover, the combination of axitinib and the cMet inhibitor crizotinib enhances survival and decreases tumor growth compared with the single treatment [65].

Despite VEGFR being widely investigated in ccRCC, PDGFR could play an important role in both tumor progression and drug resistance. The activation of PDGFR-β is associated with poor prognosis in ccRCC patients. Moreover, sunitinib-resistant ccRCC cells express low levels of the PDZ domain containing 1 (PDZK1) protein, which is unable to deactivate PDGFR-β, resulting in increased cell proliferation, tumor growth, and sunitinib insensitivity. The interaction of PDZK1 with PDGFR-β reduces the proliferation of ccRCC cells by inhibiting the PDGFR-β pathway. The induction of PDZK1 expression increases the cytotoxic properties of sunitinib in ccRCC cells [66]. PDGFR-β is also expressed in vascular pericytes that in combination with the G-protein-coupled receptor 91 (GPCR91) induce tumorigenesis and TKIs resistance-stimulating cancer stem cells (CSCs). The targeting of the PDGFR-β/GPCR91/pericytes axis inhibits CSCs and increases TKIs sensitivity in ccRCC [67]. Interestingly, the role of PDGFR-β in ccRCC is controversial, as some studies report that activation of the PDGFR-β-related pathway appears protective against both cancer progression and therapy resistance [68,69]. The explanation of this phenomenon is unknown; PDGFR-β expression could compete with VEGFR, the main angiogenesis pathway in ccRCC, or activate alternative signals/factors that lead to better therapy response. Future investigations by using patient-derived animal models might solve these discrepancies.

Data here reported emphasizes once again the ability of cancer cells to escape from cell death activating several resistance mechanisms that involve different pathways including STAT, Met, and PDGFR signals. In particular, long-term treatment with sunitinib activating these pathways leads to drug resistance causing tumor expansion with limited patient survival. Nevertheless, these studies indicate that the targeting of drug-resistance factors or the combination with different chemotherapy agents could reverse TKI resistance improving survival and quality of life of mRCC patients. Investigations of other therapeutic targets associated with drug resistance also could increase therapeutic efficacy.

Pathways involved in resistance mechanisms are schematized in Table 1 and Figure 1.

### 3.7. “Turning on and off” Autophagy

The function of autophagy in cancer is still debated. Autophagy plays a dual role in cancer, acting either as a survival mechanism that promotes tumor cell resistance under stress conditions or as a cell death pathway when excessively activated. In the context of RCC, therapeutic strategies should aim to modulate autophagy in a context-dependent manner. For instance, in cases where autophagy supports tumor cell survival and drug resistance, combining standard therapies with autophagy inhibitors may enhance treatment efficacy. Also, anticancer effects can be potentiated by coupling PI3K/mTOR inhibitors as NVP-BEZ23 and autophagy inhibitors [70]. Conversely, when autophagy leads to cell death, therapeutic approaches that promote autophagic flux could be beneficial [71]. Therefore, careful evaluation of autophagy status in RCC cells, possibly through biomarkers or functional assays, is crucial to determine whether to inhibit or stimulate the autophagic process as part of a personalized treatment strategy. Indeed, it has been reported that the expression levels of apoptosis-related genes did not show any significant association with the clinicopathological parameters of ccRCC. In contrast, elevated mRNA expression of autophagy-related genes (i.e. *ATG4*, *GABARAP*, and *p62*) was linked to earlier tumor stages, smaller tumor dimensions, and specifically for *ATG4*- and *p62*-improved disease-specific survival over five years [72]. Consistently, reduced protein levels of p62, indicative of enhanced autophagic activity, correlated with less advanced disease, fewer metastatic events, and better long-term prognosis, suggesting that in ccRCC, the activation of the autophagic machinery at the transcriptional level is associated with increased autophagic flux, occurring independently of the AMPK/mTOR pathway. Notably, ccRCC often exhibits either a monoallelic deletion or mutation of the autophagy-related gene *ATG7*, and diminished expression of autophagy-associated genes has been linked to disease progression. In line with this, ccRCC tumor tissues show decreased protein levels of both *ATG7* and *Beclin 1*, supporting the notion that in some cases autophagy functions as a tumor-suppressive mechanism during ccRCC development. Moreover, the data highlight a potential role for constitutive autophagic degradation of HIF2α by an unrecognized pathway contributing to tumor suppression [73].

Conversely, it was reported that the treatment with THZ1, a cyclin-dependent kinase 7 inhibitor, induces apoptosis and cell cycle arrest in RCC cells and acts synergistically with temsirolimus suppressing tumor growth in in vitro and in vivo models by autophagy inhibition [74]. Interestingly, the use of mTOR inhibitors, such as temsirolimus and everolimus, induces the activation of autophagy, which in turn may cause pharmacological resistance in mRCC subjects. The treatment of RCC cells with everolimus strongly induces autophagy in a dose- and time-dependent manner and the inhibition of autophagy by the treatment with chloroquine enhances everolimus-induced apoptotic cytotoxicity [75]. Autophagy activation attenuated everolimus cytotoxicity in RCC cells making cancer cells resistant to drugs; therefore, the use of mTOR inhibitors for mRCC treatment should be carefully evaluated to minimize chemotherapy resistance. To overcome this problem, phase I and II clinical trials were assessed in patients with advanced ccRCC by using a combination of everolimus and the autophagy inhibitor hydroxychloroquine [76]. No toxicity in the phase I trial after hydroxychloroquine administration was observed. The combined treatment of everolimus and hydroxychloroquine in a phase II trial reached stable disease (SD) and partial response (PR) in 67% of patients; moreover, PFS at six months of treatment was achieved in 45% of patients [76]. It was also observed that the activation of autophagy enhances sunitinib resistance through the amplification of autolysosome formation and sunitinib accumulation in autophagolysosomes leading to incomplete autophagic flux [77]. Mechanistically, sunitinib stimulates the expression of ATP-binding cassette 1 protein that induces the accumulation of this drug in autophagolysosomes. The inhibition of this protein transporter re-sensitizes mRCC cells that were resistant to sunitinib [77]. Consistently, we have demonstrated that autophagy is more activated in ccRCC cells than in normal kidney cells. The increased autophagic activity removes p53 by trapping it into autophagic vesicles leading to its degradation [78]. The inhibition of autophagy through ATG7 silencing restores p53 expression and induces cell proliferation and migration arrest, suggesting that the activation of this process promotes tumor progression via a mechanism involving p53 degradation [78]. Accordingly, it was reported that elevated levels of autophagy correlate with poorer RCC patient prognosis and the increased activity of this process is also due to VHL LOF [79]. Indeed, VHL protein directly binds to the autophagy regulator Beclin1 leading to the inhibition of autophagy initiation. The combined treatment with HIF2α and autophagy inhibitors suppresses tumor growth in ccRCC mouse models [80]. In addition, Zhu and co-workers demonstrated that the increased expression of Zinc fingers and homeoboxes protein 2 (ZHX2), a VHL substrate transcription factor with oncogenic properties, induces sunitinib resistance-activating autophagy. ZHX2 promotes angiogenesis-enhancing VEGF secretion and activates the MEK/ERK signaling pathway. Accordingly, the inhibition of autophagy by chloroquine treatment improves the anticancer effects of sunitinib confirming that this biological process contributes to drug resistance in ccRCC cells [81]. The involvement of the ERK signaling pathway in the autophagic-related drug resistance was also observed in other investigations; in fact, the activation of the autophagic pathway may reduce the cytotoxicity of mTOR inhibitors everolimus and MTI-31 in ccRCC cells by the recruitment of ERK kinases [75,82]. Pharmacological targeting of ERK by using selumetinib (AZD6244) promotes everolimus-induced cell death and reduces mTOR-dependent autophagy activation [75]. In addition, the inhibition of ERK by AZD6244 increases cell apoptosis induced by MTI-31 treatment in ccRCC cells. Moreover, the inhibition of autophagy by using chloroquine shows a similar effect to ERK inhibition, enhancing MTI-31-dependent apoptosis and thus demonstrating, as cited above, that the inhibition of mTOR causes a cytoprotective effect through the activation of autophagy [82].

Interestingly, much attention has been given to an intracellular membrane protein, paraoxonase-2 (PON2), which is closely linked to autophagy, involved in lipid metabolism, and functions to reduce oxidative stress and inflammation, leading to an improvement in mitochondrial health [83]. PON2 is highly expressed in ccRCC cells, and its silencing using vectors encoding short hairpin RNAs affects cancer-related properties such as cell proliferation, motility, and chemotherapeutic sensitivity [84]. These effects make it an interesting candidate for target therapy, perhaps developing new inhibitors through molecular docking as has been done for PON1 [85,86].

Taken together, these observations suggest that autophagy in advanced kidney carcinoma is associated with tumor progression and drug resistance by different mechanisms including ERK activation and p53 LOF/GOF. However, before attempting therapies using autophagy inhibitors, it would be appropriate to evaluate whether autophagy is protective or cancer-associated. Preclinical studies have demonstrated that the treatment with autophagy inhibitors alone or in combination with anticancer drugs reduces cancer progression. Based on these observations, several clinical trials using autophagy inhibitors combined with different chemotherapy agents have been conducted, but limited findings were published [80]. Chloroquine and hydroxychloroquine are the only FDA-approved autophagy inhibitors; treatment with these drugs was well tolerated, but further clinical studies are needed to evaluate clinical benefit. Moreover, new more efficient and specific autophagy inhibitors alone or in combination should be tested in future clinical trials.

### 3.8. Drug Removal by Transporters and Exosome Machinery

Based on kidney cell features, especially epithelial tubular cells that work by removing toxic substances and reabsorbing solutes and water, the detoxifying ability could be used by kidney cancer cells to discard drugs contributing to therapy resistance. ATP-binding cassette (ABC) transporters are expressed in different tissues including kidneys, regulate drug transport, and may be associated with multidrug resistance in solid tumors [87]. The abnormal expression of the ABC transporter (ABCB1) also called multidrug resistance (MDR) P-glycoprotein (MDR1/ABCB1) promotes the extrusion of drugs outside cancer cells causing therapy resistance in various tumors including kidney cancer [88]. It was demonstrated that ABCB1 is upregulated in different sunitinib-resistant ccRCC models and the treatment with the ABCB1 inhibitor elacridar re-sensitizes cancer cells to sunitinib, confirming the involvement of this transporter in drug resistance [89]. Sunitinib treatment might promote the expression of ABCB1 or select high-expression clones causing pharmacological resistance by drug efflux. In addition, ABCB1 activity is also increased by the synthesis of ceramide 2 in kidney tumors and chemoresistant ccRCC cells [90]. The treatment with ceramide inhibitors partially reverses chemoresistance to doxorubicin in ccRCC cells by attenuating ABCD1 transport activity [90]. Another efflux transporter named breast cancer resistance protein BCRP/ABCG2 is well characterized for its role in multidrug resistance in cancer [91]. Interestingly, the lower expression of this transporter in ccRCC tissues from untreated patients compared with normal kidney samples correlates with high tumor grade, cancer progression, and poor patient outcome. Low levels of BCRP/ABCG2 as well as MDR1/ABCB1 in cancer cells could promote tumor development, decreasing its protective effect which occurs by removing toxic and carcinogenic agents [91]. Surprisingly, this study reports that patients who express low levels of BCRP/ABCG2 exhibit decreased PFS compared with those expressing high levels of this protein after sunitinib treatment [91]. These findings are in contrast with the well-known function of BCRP/ABCG2 to contribute to multidrug resistance found in other tumors; however, data here reported regard the expression rather than the function of this transporter. Therefore, further investigations are needed to define the functional role of BCRP/ABCG2 in kidney cancer analyzing relapse and metastatic tissues of treated patients and using patient-derived xenograft mouse models. Finally, it cannot be ruled out that BCRP/ABCG2 in renal cancer might have a different role in comparison with other tumors.

Taken together, these data suggest an important role of transporters in chemotherapy resistance, since they are upregulated in drug-resistant cancer cells; thus, they might represent other possible targets to set up new mono or combined therapies for mRCC patients.

Drugs, like other molecules, can also be transported via vesicular mechanisms, including exosomes, or degraded by lysosomes. Exosomes are commonly used as drug delivery vehicles in cancer therapy; however, they also have a dark side, contributing to tumor development by influencing stromal and immune cells in the TME, and promoting cancer progression, metastasis, and therapy resistance, particularly in kidney cancer [92]. Exosome trafficking is an important mechanism of cellular communication that may enable chemotherapy-resistant cells to induce pharmacological resistance in drug-sensitive cells, thereby rendering treatment ineffective.

Long-term treatment with sunitinib promotes lysosome biosynthesis and exocytosis, thereby triggering metastasis in kidney cancer. In particular, sunitinib promotes the nuclear translocation of the TFE3 transcription factor which is involved in the lysosomal pathway leading to the activation of lysosomal exocytosis [93]. This process contributes to sunitinib resistance pumping the drug out of the cancer cell and releasing cathepsin B, which by extracellular matrix (ECM) degradation induces cell invasion and metastasis [93]. Accordingly, the treatment with the FDA-approved ketoconazole (KTZ), which has been shown to inhibit exosome biogenesis and secretion, reduces secreted exosomes in ccRCC cancer cell lines [94]. The application of KTZ in sunitinib-resistant 786-O cells restores sensitivity to sunitinib by suppressing exosome secretion, inhibiting cell proliferation and survival, and confirming the role of exosomes in chemoresistance in kidney cancer cells [94]. Another mechanism by which exosomes contribute to drug resistance is the delivery of bioactive molecules from resistant cells to adjacent sensitive cells, thereby inducing therapy insensitivity. An interesting study reports that the expression of an lncRNA (lncARSR) correlates with poor sunitinib response in ccRCC patients [95]. The lncARSR mechanism of action that leads to sunitinib resistance involves the activation of AXL and c-MET oncogenes and confers sunitinib resistance to sensitive cells by transporting this lncRNA, which sequesters miR-34 and miR-449, through the exosome machinery [95]. The treatment of sunitinib-resistant RCC cells and xenografts models with specific nucleic acid sequences targeting lncARSR or inhibitors of AXL/c-MET recovers sunitinib sensitivity [95,96]. KTZ not only acts on lncARSR, but also inhibits exosome biogenesis, the formation of intraluminal vesicles as well as vesicle transport through cytoskeletal filaments, by affecting Alix, Neutral sphingomyelinase 2 (nSMase2), and Rab27a [92]. Particular attention is given to proteins involved in secretion processes. Among these, RAB27B is upregulated in RCC cells, especially in sunitinib-resistant ones, whose sensitivity is restored by RAB27B knockout [97]. RNA sequencing and pathway analysis indicated that the function of RAB27B may be mediated by the MAPK and VEGF signaling pathways.

Targeting exosome synthesis and trafficking represents a promising novel approach for mRCC therapy. In this context, the combined use of Nexinhib20 and GW4869 with cisplatin or etoposide drugs already employed in first-line chemotherapy for small-cell lung cancer could also be applied to RCC. This combination aims to inhibit RAB27A and nSMase2, an enzyme involved in ceramide generation. Additional general inhibitors that could be tested include calpeptin, manumycin A, Y-27632, D-pantethine, and imipramine [98]. Furthermore, exosomes can be exploited as delivery vehicles for interfering RNAs or mimic molecules to counteract the effects of oncogenic ncRNAs or enhance the activity of tumor-suppressive ncRNAs involved in the pathways described below.

Other miRNAs secreted by EVs, such as miR-27a targeting SFRP1, can sustain angiogenesis, as it is known that pro-angiogenic miRNAs confer resistance following doxorubicin treatment [99]. TKI- and sorafenib-resistant RCC cells display low exosomal miR-549a levels, leading to increased HIF1α expression, which promotes VEGF secretion, angiogenesis, vascular permeability, and cell migration. This process is further sustained by the activation of the positive feedback VEGFR2–ERK–XPO5 pathway [100].

Although extracellular vesicles (EVs) play a pivotal role in transferring resistance by diffusing their content (proteins, lipids, and nucleic acids) both within the TME and through biological fluids, EV-based therapies are also particularly suitable for drug delivery and therapeutic applications [92]. For example, EVs with low levels of miR-30c-5p, which targets heat-shock protein (HSP) 5, were detected in ccRCC patients, and increased levels of this miRNA affect cancer progression [101]. The influence of exosomes is primarily relevant to cancer-associated fibroblasts (CAFs) in the TME, which are increased in metastatic ccRCC [92]. In this context, CAFs secrete exosomes containing miR-224-5p, which is internalized by ccRCC cells, promoting the migration of metastatic features. Similarly, CAF-derived miR-181d-5p affects RNF43, activating the Wnt/β-catenin signaling pathway. On the other hand, ccRCC cells can release exosomes containing circSAFB2, which interferes with the miR-620/JAK1/STAT3 axis promoting polarization of M2 macrophages [92].

Another strategic use of exosomes is applied to immunotherapy and carried out by developing immunotherapeutic vaccines using exosomes derived from RCC cells that induced cytotoxic T lymphocytes against RCC antigens resulting in improved anticancer properties. Other similar approaches were recently used to generate vaccines by using cancer-derived exosomes, which showed increased survival in immunized RCC mice [92].

Although these data support the development of promising future mono or combination therapies aimed at improving overall survival (OS) in patients with advanced ccRCC, other aspects must be considered, particularly the role of EVs in the stimulation of angiogenesis, immune evasion, and tumor progression. Indeed, EVs secreted by RCC cells can inhibit T cell proliferation, natural killer cell activation, and dendritic cell maturation, ultimately leading to reduced sensitivity to ICI therapy [102]. This is partly due to the presence of PD-L1 on the surface of RCC-derived EVs, as well as their ability to induce PD-L1 expression and modulate the immune system [92]. For example, vesicles with miR-224-5p have been shown to promote PD-L1 expression in RCC cells, thereby enhancing resistance to T cell-mediated cytotoxicity [103].

In light of these data, the targeting of exosome machinery could be a new encouraging strategy to treat metastatic kidney cancer.

### 3.9. Ferroptosis Inhibition

Ferroptosis is another type of cell death associated with iron and reactive oxygen species (ROS); in particular, ferroptosis is correlated with the oxidation of polyunsaturated fatty acids, iron metabolism, and glutathione peroxidase 4 (GPX4) inactivation. Different cell death pathways, including ferroptosis, seem to affect resistance to targeted therapy, and the stimulation of this process could reverse drug resistance in tumors including kidney cancer [104,105]. Chen and colleagues report that resistance to TKIs in kidney cancer may be associated with signaling pathways, which lead to ferroptosis inactivation. They detected that in ccRCC tissues and cells resistant to TKI, interleukin 6 (IL6) functions as an upstream element which stimulates the expression of the solute carrier family 7 member 11 (SLC7A11) via JAK2-STAT3 signaling leading to ferroptosis inhibition and TKI resistance. The application of Erastin, a ferroptosis activator, seems to reverse the IL6 effect re-sensitizing cancer cells to pharmacological treatment [106]. A similar mechanism is driven by the tribbles pseudokinase 3 (TRIB3), an oncoprotein involved in tumor drug resistance and ferroptosis regulation that positively modulates the expression of SLC7A11. The silencing of TRIB3 inhibits cell proliferation and migration in ccRCC cells and activates ferroptosis negatively affecting the expression of the SLC7A11/GPX4 pathway. Moreover, TRIB3 knockdown enhances the therapeutic efficacy of sunitinib in ccRCC cells triggering sunitinib-induced ferroptosis [107]. Ferroptosis in ccRCC cells may also be modulated by the absence of melanoma 2 (AIM2), a receptor protein implicated in cancer progression and chemotherapy resistance. The overexpression of AIM2 promotes RCC progression and sunitinib resistance by FOXO3a phosphorylation and proteasome degradation, leading to the reduction in Acyl-CoA Synthetase long chain family member 4 and ferroptosis inhibition [108]. Moreover, the expression of SLC7A11 is regulated by nuclear factor erythroid 2-related factor 2 (NRF2), a transcription factor that is often hyperactivated in many tumors, playing a key role in cancer progression, metastasis, and therapy resistance. In ccRCC, the constitutive activity of NRF2 and the following constitutive overexpression of the genes under its transcriptional control is triggered by ROS, determining the switch from a protective to tumorigenic profile in an attempt to limit oxidative stress and inflammation [109]. This condition is exacerbated by drugs such as cisplatin, increasing the activation of the NRF2/ARE pathway and mutations in the genes *KEAP1* and *CUL3*, which in normal conditions lead to NRF2 inactivation by ubiquitin [109]. Several genes are regulated by this transcription factor and engaged in the detoxification processes; many of them are involved also in ferroptosis. In addition to SLC7A11, we found glutathione S-transferases and heme oxygenase- (HO-1)1, to mention just a few.

NRF2 stabilization depends on the expression of dipeptidyl peptidase 9 (DPP9), which is upregulated in ccRCC and correlates with advanced tumor stage and poor prognosis. DPP9 overexpression inhibits ferroptosis and induces sorafenib resistance in ccRCC cells, in a mechanism involving the activation of the NRF2/SLC7A11 axis [110].

We underline that NRF2 represents a crucial node interconnected with the deregulated PI3K/Akt pathway, which modulates the expression of HO-1 and autophagy, regulating the expression of SQSTM1 [109]; on the other hand, mutations in the NRF2 promoter upregulating it were frequent in patients with metastatic ccRCC and not responding to VEGF-targeted therapy [111].

It is reasonable to assume that promoting the inhibition of this factor induces anticancer effects. Indeed, it has been shown that the miR-200a-3p/141-3p/KEAP1 axis promotes the proteasomal degradation of NFE2 [109]. Decreased levels of miR-200 were observed in RCC patients with inactive fumarase, known to be associated with accumulation of fumarate affecting KEAP1 and NRF2 increase.

Since NRF2 inhibitors have proven ineffective, modulating it by targeting its regulatory pathways could be considered. Among the possible candidates, we mention dimethyl fumarate and KEAP1 knockdown/silencing, which may ameliorate tumor sensitivity in axitinib- or sunitinib-resistant patients or in association with the VEGFR inhibitor pazopanib. Interestingly, anticancer properties demonstrated by the ginsenoside Rh4, increasing ferroptosis in RCC, appear mediated by mechanisms involving mainly NRF2 [112].

Kidney cancer cells may reduce ferroptosis by utilizing alternative mechanisms, such as the expression of ncRNAs that promote cancer progression and are resistant to therapy. The overexpression of the lncRNA STX17-DT induces axitinib resistance in ccRCC cells and is linked to poor prognosis in ccRCC patients. STX17-DT upregulation causes the decrease in mitochondrial ROS attenuating ferroptosis. STX17-DT is packaged into extracellular vesicles and thus may transmit axitinib resistance to surrounding cells. Silencing of STX17-DT enhances the therapeutic efficacy of axitinib in xenograft models of ccRCC by increasing ferroptosis [113].

Findings reported here suggest that chemotherapeutic treatment induces cell death by ferroptosis; however, cancer cells adopt different mechanisms to prevent ferroptosis activation, causing resistance to pharmacological treatment. Therefore, molecules that can activate ferroptosis or target ferroptosis inhibitors may enhance the anticancer effects of conventional drugs, helping to overcome therapy resistance. Nevertheless, cell death induced by ferroptosis activation could lead to the release of damage-associated molecular patterns by tumor cells, which may activate the immune system, enhancing anticancer therapy or contributing to drug resistance. It has been reported that chemotherapy-induced activation of apoptosis can lead to the production of molecules, such as phosphatidylserine, which in turn promote the accumulation of M2-like macrophages. M2 TAMs may affect the outcome of tumors by inhibiting T cell-mediated anti-tumor immune response, causing tumor progression and resistance to anticancer drugs [114]. Accordingly, it was reported that the activation of ferroptosis may also induce the release of molecules that trigger the STING-dependent DNA sensor pathway, enhancing the infiltration of M2-like macrophages that facilitate tumor progression in pancreatic carcinoma [115].

According to these observations, the use of agents that induce ferroptosis for cancer treatment should be carefully evaluated to neutralize factors that activate M2-like macrophages, which contribute to immunosuppression.

These topics are schematized in Figure 2.

### 3.10. Mechanisms Engaging lncRNAs and circRNAs in the Chemoresistance

The tumor heterogeneity of RCC drives the selection of clones by drug treatments and is mainly associated with genomic instability. In this context, a pan-study identified 170 ncRNAs that characterize signature profiles; among them, four (LINC00460, AC073218.1, AC010789.1, and COLCA1) correlate with clinical outcomes [116]. This finding reveals the pivotal role of ncRNAs in the development of drug resistance. In addition, specific SNPs occurring in genes coding ncRNAs are associated with drug resistance, such as LncARSR affecting sunitinib response, and especially some variants (rs7859384) which can be tightly correlated [117]. Others, like GPRC5D-AS1, are correlated to migration or invasion, modulating EMT markers, but the mechanisms are not already defined [118].

Sunitinib is one of the drugs on which the investigations concerning resistance are most numerous, also extending to in-depth study of the role of ncRNAs. In the literature, most studies describe the action of ncRNAs as serving as sponges for microRNAs, leading to the modulation of target gene expression. This chemotherapy agent dramatically increased one of the more investigated lncRNAs, MALAT1, and its knockdown reverted chemoresistance. MALAT1 acts by sponging miR-362-3p, upregulating G3BP1 and heterogeneous nuclear RNA-binding proteins, which interfere with the signaling cascade by binding Ras-GTPase [119]. Concerning sunitinib resistance in ccRCC, upregulation of LINC00667 affected miR-143-3p functioning as a sponge, also determining an increase in ZEB1, sustaining aggressive tumor features, such as EMT and migration in xenograft mice [120]. Another lncRNA with a similar function is FAM13A-AS1. This decoys miR-141-3p, which typically targets NEK6, a kinase driving cell division, and its consequent increase promotes tumorigenesis [121]. Other mechanisms involve the interaction of ncRNAs with proteins, interfering with their stabilization. For example, the interaction of SNHG12 with SP1 prevents its ubiquitylation-mediated degradation, without affecting its function as a transcription factor and, consequently, maintaining the expression of genes such as CDCA3, an essential regulator of cell mitosis [122]. In ccRCC, increased expression of IGFL2-AS1 following sunitinib treatment is associated with higher tumor growth and patient mortality [123]. The role of this ncRNA affects the splicing of TP53INP2, encoding a protein with dual regulatory properties that triggers autophagosome formation and also functions as a transcription factor. Altered TP53INP2 transcripts appear to be more efficiently translated on high molecular weight polysomes. This effect is mediated by IGFL2-AS1 competition with heterogeneous nuclear ribonucleoprotein C (hnRNPC), which promotes alternative cleavage and polyadenylation, along with other functions such as nuclear export, favored by its m6 A modification. Interestingly, IGFL2-AS1 may be packaged into extracellular vesicles by hnRNPC, transmitting sunitinib resistance to surrounding sensitive cells. IGFL2-AS1 knockdown highlights its therapeutic potential to sensitize cells to apoptosis and reverse sunitinib resistance both in vitro and in vivo [123].

More recently, the involvement of other types of ncRNAs, such as circRNAs, has also been studied. CircSNX6 is overexpressed in a cohort of 81 patients [124], which can act as a sponge of microRNAs, mainly like lncRNAs. CircSNX6 affects miR-1184, leading to higher levels of intracellular lysophosphatidic acid, which enhances glycerophosphocholine phosphodiesterase 1 (GPCPD1) expression and induces sunitinib resistance. Instead, the downregulation of circEHD2 occurs in sunitinib-sensitive RCCs, enhances the miR-4731-5p/ABCF2 axis, and its inhibition reverses drug resistance [125]. Unlike upregulated circAGAP1, suppressing miR-149-5p, miR-455-5p, and miR-15a-5p regulates the expression of sunitinib target PDGFR and inhibits proliferation and migration in sunitinib-sensitive ccRCC cells [126].

Sorafenib resistance also involves the deregulation of ncRNAs. Among the resistance processes activated by sorafenib, an increase in the levels of KIF9-AS1 has been observed, which via miR-497-5p decreases apoptosis by deregulating TGF-β and autophagy [127]. Another lncRNA upregulated by sorafenib is PLK1S1, also associated with poor survival; it sequesters miR-653 by decreasing the expression of the downstream CXCR5 target [128]. The activation of inflammation to escape sorafenib sensitivity constitutes a pathway modulated by other lncRNAs, such as lncRNA-SRLR, that directly binds to NF-κB and stimulates the expression and activation of IL-6 and STAT3 [129].

Additionally, chemoresistance phenomena have been reported for 5-Fu treatment, correlated with low expression of ADAMTS9-AS2 lncRNA, which promotes an increase in FOXO1 sponging miR-27a-3p [130].

The information reported above is schematized in the diagrams of Figure 3 and summarized in Table 2.

The limited data regarding the study of lncRNAs and cirRNAs are an indication of how little knowledge there is on the regulatory functions of these molecules in RCC.

Overall, data here reported indicate that cancer cells adopt many strategies to evade cell death leading to drug insensitivity and cancer progression. Despite great progress obtained, there is still much work to do in order to set up specific therapies capable of eluding pharmacological resistance. Most promised studies were mainly carried out in in vitro and in pre-clinical animal models limiting therapeutic availability in a short time. Future clinical studies should be performed to test the efficacy of therapeutic targets that yielded positive responses in pre-clinical investigations.

## 4. Conclusions

In recent years, understanding the molecular mechanisms behind resistance has opened new avenues for addressing the issue. Pharmacological insensitivity of cancer cells leads to the failure of current therapies with limited overall survival for mRCC patients. Genetic mutations activating oncogenic signals or inhibiting tumor suppressor factors are the main causes of resistance after prolonged treatment. Other players such as cell adaptation to changes in TME, alteration of cellular transport as well as the activation or the shutdown of different biological processes including EMT, autophagy, and ferroptosis are also important. Patients who have failed first-line therapy should benefit from targeted more efficient second-line treatments to improve their survival and quality of life. Therefore, new investigations should be carried out to understand resistance mechanisms to detect new therapeutic targets that could overcome this problem. NEXT-generation drugs targeting patient-specific cancer factors and multitarget therapies could prevent drug resistance. New strategies addressed to the activation of immunity response against tumor cells could also enhance the benefits of anticancer treatments. If these approaches are effective, they could open a new era for the treatment of metastatic renal carcinoma and other cancers resistant to conventional therapies.

## Figures and Tables

**Figure 1 cancers-17-01613-f001:**
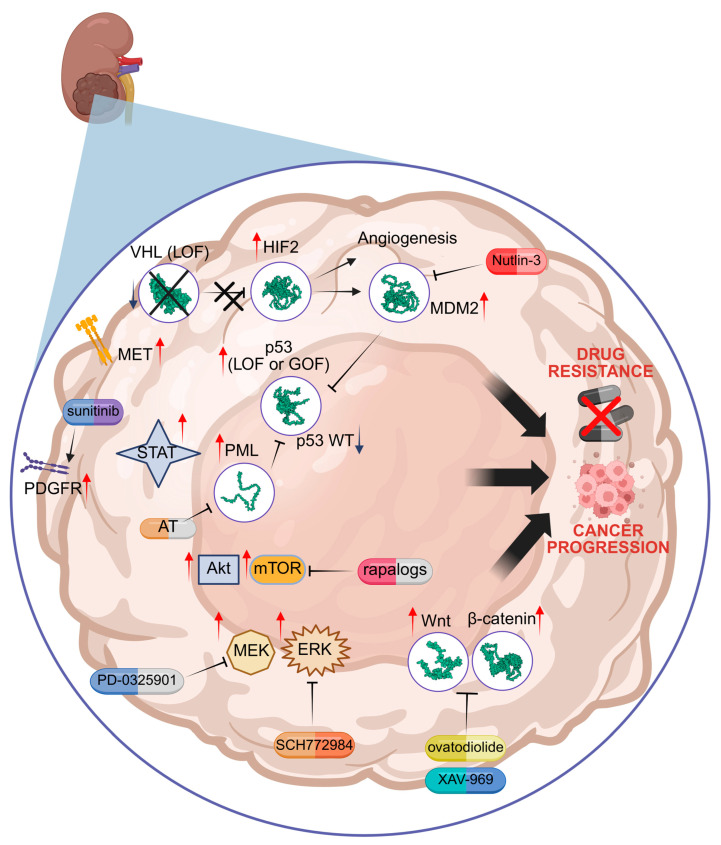
Molecular mechanisms of drug resistance linked to HIF2, p53 mutation, PML, mTOR, MEK/ERK, and Wnt/β-catenin. Specific inhibitors are indicated in red. VHL = von Hippel–Lindau; HIF = hypoxia inducible factor; MDM2 = Murine Double Minute 2; PML = Promyelocytic Leukaemia protein; AT = arsenic trioxide; LOF = loss of function; GOF = gain of function; mTOR = mammalian target of rapamycin; MEK = mitogen-activated protein kinase kinase; ERK = extracellular signal-regulated kinase; Wnt = wingless-related integration site; PDGFR = platelet-derived growth factor receptor; MET = mesenchymal epithelial transition factor receptor; STAT = signal transducer and activator of transcription. Red arrows = increased activity; Blue arrow = decreased activity. Illustration created by BioRender (https://app.biorender.com/; last login: 5 May 2025).

**Figure 2 cancers-17-01613-f002:**
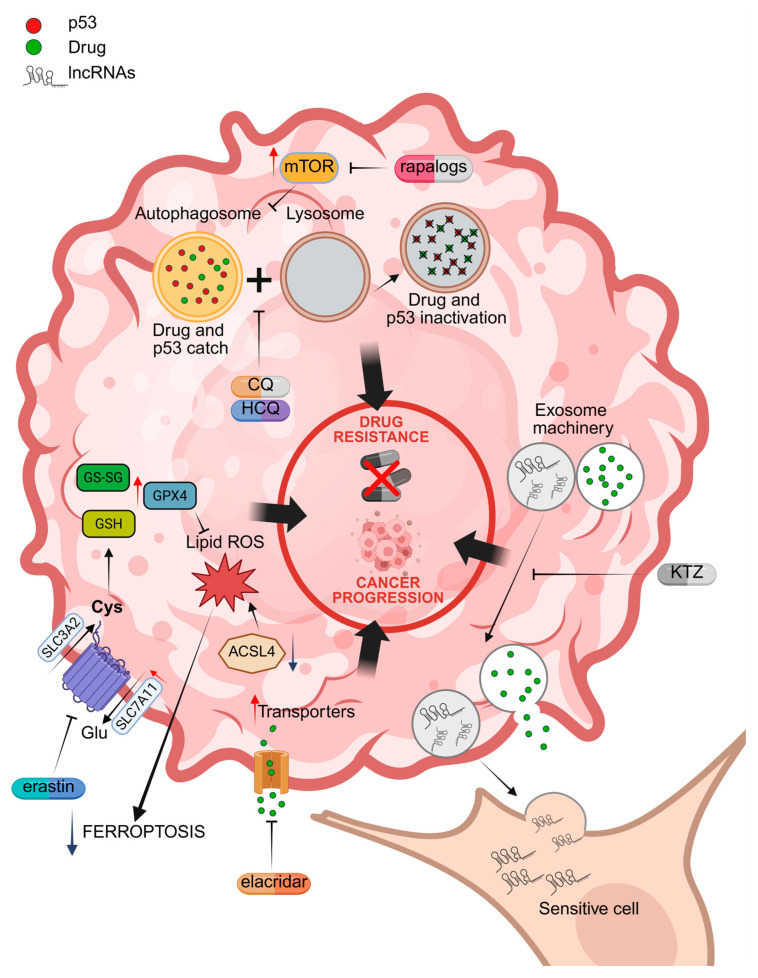
Molecular mechanisms of drug resistance associated with the activation of autophagy, drug extrusion channels, exosome machinery, and the inhibition of ferroptosis. Specific inhibitors are indicated in red. CQ = chloroquine; HCQ = hydroxychloroquine; KTZ = ketoconazole; lncRNA = oncogenic lncRNAs; GPX4 = glutathione peroxidase 4; ACSL4 = Acyl CoA synthetase long chain family member 4; SLC7A11 = solute carrier family 7 member 11; GSH = reduced glutathione; GS-SG = oxidized glutathione. Red arrows = increased activity; Blue arrows = reduced activity. Illustration created by BioRender (https://app.biorender.com/; last login: 29 April 2025).

**Figure 3 cancers-17-01613-f003:**
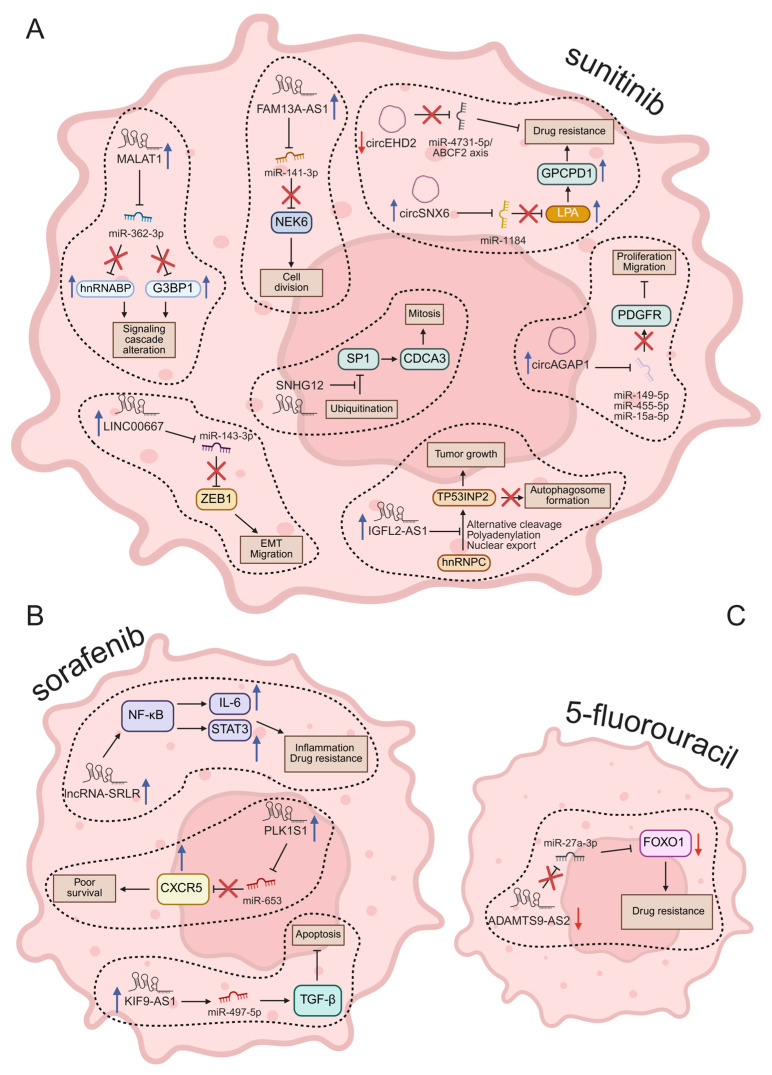
Molecular mechanisms regulated by ncRNAs in drug resistance of RCC. In (**A**), after sunitinib treatment; in (**B**), changed ncRNA by sorafenib; in (**C**), ncRNA modulated by 5-Fu. Red arrows = downregulated; blue arrows = upregulated. Illustration created by BioRender (https://app.biorender.com/; last login: 29 April 2025).

**Table 1 cancers-17-01613-t001:** Signaling pathways involved in drug resistance and related mechanisms in ccRCC cells.

Pathway	Drug	Resistance Mechanism	Alternative Target/Drugs
HIF	PT2385, PT2399	Gatekeeper mutation (G323E) in HIF2α; mutations in HIF1α and HIF2α binding site	S1P; FTY720
HIF	sunitinib	Plk1 activation	Plk1
HIF	sunitinib	ETS1 deacetylation	HDAC8
HIF	sunitinib	VEGFA increase	CHD1L
p53	chemotherapy/rapamycin	MDM2/Hdm2-mediated p53 suppression	Hdm2; HIF2α
p53	sorafenib/sunitinib	MDM2 activation	MDM2; nutlin-3
p53	adriamycin/sunitinib	HIF activation	HIF
p53	chemotherapy	PML expression	PML; arsenic trioxide
mTOR	rapalogs	mTORC2/Akt	ERK, STAT3
mTOR	rapalogs	mTORC1 via TME	TME factors
mTOR	TKIs	AGK activation	PI3K–Akt–mTOR
mTOR	rapalink-1	not yet observed	
mTOR	sapanisertib/cabozantinib	not yet observed	
MAP2K2	sunitinib/axitinib/sorafenib	MEK/ERK activation	SP1
MAPK/AP-1	TKIs	IFITM3	IFITM3
Raf/MEK/ERK	sorafenib/sunitinib	SOX9 expression	SOX9
MAPK	sunitinib	MDSC accumulation	MEK; PD-0325901
ERK	sunitinib	MAGI3 downregulation	Ang-(1-7)/MAS/ERK
Akt/ERK	sorafenib	PGK1 upregulation	C-X-C chemokine CXCR4
Wnt/β-catenin	sorafenib/sunitinib	β-catenin phosphorylation	β-catenin; ovatodiolide
CXCR4 and β-catenin	sunitinib	BIRC6 overexpression	Wnt/β-catenin; XAV-939
Wnt3a/FZD1	imatinib	PDIA6 upregulation	PDIA6
STAT1	radiotherapy/Taxol	STAT1 overexpression	
STAT2	pazopanib	SLC27A3 increase	SLC27A3
STAT3	sunitinib	circular RNA circPTPN12	
STAT3	pazopanib	Gankyrin	Gankyrin/CCR3
cMet	sorafenib/sunitinib	IRAK1 receptor activation	IRAK1
MET/AXL	sunitinib	MET/AXL activation	cabozantinib
cMet	sunitinib	cMet activation	axitinib; crizotinib
PDGFR-β	sunitinib	PDZK1 protein downregulation	PDZK1 induction
PDGFR-β	TKIs	GPCR91 activation	PDGFR-β/GPCR91

**Table 2 cancers-17-01613-t002:** Networks of lncRNAs or cirRNAs and their downstream targets.

ncRNA	Drug	Targeted microRNAs	Downstream Genes
MALAT1	sunitinib	miR-362-3p	*G3BP1*
LINC00667	sunitinib	miR-143-3p	*ZEB1*
FAM13A-AS1	sunitinib	miR-141-3p	*NEK6*
SNHG12	sunitinib	SP1	*CDCA3*
IGFL2-AS1	sunitinib	hnRNPC	*TP53INP2*
circSNX6	sunitinib	miR-1184	*GPCPD1*
circEHD2	sunitinib	miR-4731-5p	*ABCF2*
circAGAP1	sunitinib	miR-149-5p, miR-455-5p, miR-15a-5p	*PDGFR*
KIF9-AS1	sorafenib	miR-497-5p	*TGF-β*
PLK1S1	sorafenib	miR-653	*CXCR5*
lncRNA-SRLR	sorafenib	NF-κB	*IL-6*, *STAT3*
ADAMTS9-AS2	5-Fu	miR-27a-3p	*FOXO1*

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
