# Peer review of "Molecular Mechanisms of Drug Resistance in Clear Cell Renal Cell Carcinoma"

_cancers, 2025, doi:10.3390/cancers17101613_

Round 1
Reviewer 1 Report
Comments and Suggestions for Authors
The manuscript “Molecular Mechanisms of Drug Resistance in Kidney Carcinoma” is a review article regarding the resistance mechanisms involved in Renal cell carcinoma. The review design is average and the manuscript may be of interest for the readers. However, some important concerns prevent the publication of the manuscript in the current form:
- The manuscript requires a native English-speaker revision. Some expressions are unacceptable e.g “The discussed pathways are schematize in Figure 1.”
- Both simple summary and the abstract structure could be improved providing a clearer landscape of the background and the aim of the review
- It is not clear if authors focus on renal cancer, metastatic renal cancer or clear cell renal cell carcinoma only.
- Authors should state what is the contribution of their review to similar recent reviews already published.
- The authors somehow ignored part of the available literature on this topic. Clearly, it is not satisfying that a such broad topic includes only 83 references. For instance, a recent article that revealed that the enzyme Paraoxonase-2 is involved in clear cell renal cell carcinoma proliferation and resistance to chemotherapeutics (PMID: 38706121). A comprehensive review regarding the resistance mechanisms involved in Renal cell carcinoma should include these observations.
- Authors should expand the discussion on the reported studies, underlying strong points and limitations for the reported studies.
- In line 497 authors introduce Nrf2 which is a known key factor for therapy resistance in renal cell carcinoma (PMID: 39769005). This topic requires a larger discussion.
- Figure must be improved since are excessively blurry.
- There are strange symbols in the text.
- The conclusion section should be totally rewritten since it actually does not add anything as it is now. Please clearly focus on actual state-of-art and future perspectives.
The manuscript requires a native English-speaker revision.
Author Response
REPLY to the Reviewer 1
The manuscript “Molecular Mechanisms of Drug Resistance in Kidney Carcinoma” is a review article regarding the resistance mechanisms involved in Renal cell carcinoma. The review design is average and the manuscript may be of interest for the readers. However, some important concerns prevent the publication of the manuscript in the current form:
- The manuscript requires a native English-speaker revision. Some expressions are unacceptable e.g. “The discussed pathways are schematized in Figure 1.”
Answer: We apologize. We have made an English revision to the manuscript, and we hope the new version meets the required standards.
- Both simple summary and the abstract structure could be improved providing a clearer landscape of the background and the aim of the review
Answer: As suggested, we modified the simple summary and abstract, including new phrases regarding background and aim. The changes are marked in red:
“Simple Summary: Metastatic renal carcinoma (mRCC) is very hard to cure, despite the use of new-generation therapies that include inhibitors of key proteins such as tyrosine kinases (TK) and blockers that stimulate the immune response against cancer cells called immune checkpoint inhibitors (ICIs). The efficacy of chemotherapy decreases over time because of innate and acquired resistance that dramatically reduces the life expectancy of mRCC patients. This review aims to analyze the molecular mechanisms involved in cancer drug resistance in order to identify new therapeutic targets that could overcome this matter. We discussed the main pathways involved in pharmacological resistance including angiogenic factors, oncogenes, tumor suppressor proteins, vesicular transport, ferroptosis, and non-coding RNAs. Clinical trial studies targeting factors involved in resistance or multi-target therapies are also reported. The results discussed could suggest the setting of specific treatments with improvements for subjects suffering from mRCC and could help clinicians choose the best options for the management of ccRCC patients.”
“Abstract: Renal cell carcinoma (RCC) accounts for about 3% of all human tumors. Alterations of oxygen, lipids, iron, and energy metabolism are involved in carcinogenesis, development, and expansion. Thirty % of patients affected by clear cell renal cell carcinoma (ccRCC) will develop relapses or distance metastases (mRCC), dramatically reducing their life expectancy. Current first-line therapies for mRCC patients are based on treatment with immune checkpoint inhibitors (ICIs) alone and in combination with each other or with tyrosine kinase inhibitors (TKIs). However, only 20% of patients show a mild response because of innate or acquired drug resistance during long-term treatment; therefore, resistant patients need alternative first-line or second-line therapies. Pharmacological resistance represents a big problem that counteracts the efficacy of treatment by reducing overall survival (OS) in mRCC patients. Investigating the molecular mechanisms underlying drug resistance is crucial to overcoming drug insensitivity and enhancing therapeutic outcomes. In this review, we emphasize the latest and most significant studies on the molecular mechanisms that drive drug resistance in ccRCC carcinoma. Particular attention is given to the key signaling pathways involved in resistance, including those mediated by HIF, p53, Akt-mTOR, MEK-ERK cascades, Wnt signaling, autophagy, membrane transporters, ferroptosis, and non-coding RNAs. Understanding these resistance mechanisms is essential for developing new therapeutic strategies aimed to enhancing overall OS and improving the quality of life for mRCC patients. The review also discusses recent clinical trial findings on the use of specific inhibitors able to circumvent drug resistance. The data presented here could be valuable for clinicians in understanding the mechanisms of drug resistance, ultimately aiding in the management of ccRCC patients.”
- It is not clear if authors focus on renal cancer, metastatic renal cancer or clear cell renal cell carcinoma only.
Answer: This work is based on data from clear cell renal cell carcinoma (ccRCC) and its metastatic forms (mRCC). The other histotypes were not considered. To clarify this point, we have modified the title as follows: “Molecular Mechanisms of Drug Resistance in Clear Cell Renal Cell Carcinoma”. Wherever possible, the acronym RCC was changed in ccRCC.
- Authors should state what is the contribution of their review to similar recent reviews already published.
Answer: Following the Reviewer’s criticisms we have inserted a new paragraph at the end of the Introduction explaining our contribution to this issue. The new paragraph is described below:
In this review, we present the latest findings on the molecular mechanisms of drug resistance employed by cancer cells to evade chemotherapy-induced cell death, as there has been limited discussion on this topic. Cancer cells use different ways to elude cell death during prolonged treatment with anticancer therapies. Drug resistance mechanisms include the enhancement of anti-vascular and survival pathways, cell adaptation to the harmful microenvironment, and epithelial-to-mesenchymal transition (EMT). We have thorough these aspects with particular emphasis on signaling pathways, biological processes, and non-coding RNA (ncRNAs) ex-pression involved in drug insensitivity. NEXT-generation drugs and multitarget therapies designed to bypass resistance mechanisms, along with clinical trial results, are also dis-cussed. Mechanisms of resistance related to inflammation and the immune system have been extensively covered in the literature and, therefore, are not included in this work.”
- The authors somehow ignored part of the available literature on this topic. Clearly, it is not satisfying that a such broad topic includes only 83 references. For instance, a recent article that revealed that the enzyme Paraoxonase-2 is involved in clear cell renal cell carcinoma proliferation and resistance to chemotherapeutics (PMID: 38706121). A comprehensive review regarding the resistance mechanisms involved in Renal cell carcinoma should include these observations.
Answer: The reviewer is right in pointing out that the number of references was limited for such a broad topic. We underline that we considered as many as 583 manuscripts for this review. However, we would like to point out that many works currently available in the literature lack consistent and clear data or tend to repeat concepts already covered in our review. We have therefore added several more references to the revised manuscript, with particular care to include the most recent and relevant studies.
Concerning PON2, it is true that it represents a potential therapeutic target; therefore, we have added the following sentence to the revised manuscript:
“Interestingly, much attention has been given to an intracellular membrane protein, paraoxonase-2 (PON2), which is closely linked to autophagy, involved in lipid metabolism, and functions to reduce oxidative stress and inflammation, leading to an improvement in mitochondrial health [83]. PON2 is highly expressed in ccRCC cells, and its silencing using vectors encoding short hairpin RNAs affects cancer-related properties such as cell proliferation, motility, and chemotherapeutic sensitivity [84]. These effects make it an interesting candidate for target therapy, perhaps developing new inhibitors through molecular docking as has been done for PON1 [85,86].”
Moreover, to enhance information regarding this topic a new chapter entitled “Other pathways” was inserted in the text:
"3.6. Other pathways
Mechanisms of drug resistance involve other signaling pathways including STAT, mesenchymal-epithelial transition factor (cMet), and platelet-derived growth factor receptor (PDGFR). STAT1 is overexpressed in ccRCC tissues and contributes to chemotherapy resistance. The suppression of STAT1 inhibits cell growth in both in vitro and in vivo models for ccRCC. Moreover, the inhibition of STAT1 re-sensitizes ccRCC cells to radiotherapy and Taxol treatment [59]. Furthermore, Lu D. and colleagues report that STAT2 promotes the expression of the solute carrier family 27 member 3 (SLC27A3), involved in Pazopanib resistance in ccRCC cells. SLC27A3 knockdown overcomes Pazopanib resistance, therefore SLC27A3 and STAT2 could represent new therapeutic targets for mRCC treatment [60]. Also, activation of the STAT3 pathway contributes to sunitinib resistance in ccRCC; in fact, it was reported that the upregulation of the circular RNA circPTPN12 in ccRCC tissues was associated with poor prognosis. Mechanistically, the overexpression of circPTPN12 induces the activation of the STAT3 signaling pathway enhancing proliferation, migration, invasion, and sunitinib resistance [61]. In ccRCC tissues, STAT3 may be activated by the overexpression of gankyrin leading to the upregu-lation of the C-C motif chemokine ligand 24 (CCL24) that generates a positive loop en-hancing the expression of gankyrin and STAT3 via C-C chemokine receptor type 3 (CCR3). Increased levels of gankyrin stimulate proliferation, invasion, migration, and Pazopanib resistance. Gankyrin knockdown or CCR3 inhibition reverses the resistance to Pazopanib and prevents metastasis in “in vivo” models for ccRCC [62].
Another mechanism of TKI resistance involves the upregulation of the oncogene cMet. As already reported, TKIs were widely used for systemic therapy in mRCC patients; however, pharmacological resistance phenomena following sunitinib and sorafenib treatment were observed. In particular, resistant ccRCC cells exhibit the activation of the cMet receptor IRAK1 and the reduction of the tumor suppressor MCPIP1 expression, triggering the metastatic process [63]. Moreover, the prolonged treatment with sunitinib enhances the expression of both MET and AXL metastasis-related receptors in ccRCC cell lines increasing cell migration and invasion. The inhibition of AXL and MET treating cells with the multi-kinase inhibitor cabozantinib overcomes resistance due to long-term treatment with sunitinib in ccRCC cells [64]. Another study reports that the prolonged treatment with sunitinib in a patient-derived ccRCC xenograft model causes the increased expression of cMet factor, which is involved in acquired drug resistance. The treatment with axitinib, a well-tolerated tyrosine inhibitor, reduces tumor growth in sunitinib-resistant ccRCC tis-sues. Moreover, the combination of axitinib and the cMet inhibitor crizotinib enhances survival and decreases tumor growth compared with the single treatment [65].
Despite VEGFR being widely investigated in ccRCC, PDGFR could play an important role in both tumor progression and drug resistance. The activation of PDGFR-β is associ-ated with poor prognosis in ccRCC patients. Moreover, sunitinib-resistant ccRCC cells ex-press low levels of the PDZ domain containing 1 (PDZK1) protein, which is unable to de-activate PDGFR-β, resulting in increased cell proliferation, tumor growth, and sunitinib insensitivity. The interaction of PDZK1 with PDGFR-β reduces the proliferation of ccRCC cells by inhibiting the PDGFR-β pathway. The induction of PDZK1 expression increases the cytotoxic properties of sunitinib in ccRCC cells [66]. PDGFR-β is also expressed in vascular pericytes that in combination with the G-protein-coupled receptor 91 (GPCR91) induces tumorigenesis and TKIs resistance stimulating cancer stem cells (CSCs). The tar-geting of the PDGFR-β/ GPCR91/pericytes axis inhibits CSCs and increases TKIs sensitiv-ity in ccRCC [67]. Interestingly, the role of PDGFR-β in ccRCC is controversial, as some studies report that activation of the PDGFR-β-related pathway appears protective against both cancer progression and therapy resistance [68, 69]. The explanation of this phenomenon is unknown; PDGFR-β expression could compete with VEGFR, the main angiogenesis pathway in ccRCC, or activate alternative signals/factors that lead to better therapy response. Future investigations by using patient-derived animal models might solve these discrepancies.
Data here reported emphasizing once again the ability of cancer cells to escape from cell death activating several resistance mechanisms that involve different pathways including STAT, Met, and PDGFR signals. In particular, long-term treatment with sunitinib activating these pathways leads to drug resistance causing tumor expansion with limited patient survival. Nevertheless, these studies indicate that the targeting of drug-resistance factors or the combination with different chemotherapy agents could reverse TKI resistance improving survival and quality of life of mRCC patients. Investigations of other therapeutic targets associated with drug resistance also could increase therapeutic efficacy.”
- Authors should expand the discussion on the reported studies, underlying strong points and limitations for the reported studies.
Answer: We agree with the Reviewer that the Discussion can be expanded and improved. Changes were reported below:
Chapter 3.1: Clinical trials by using HIF inhibitors were included:
“Some HIF2α inhibitors, such as DFF332, require further investigation to establish a recommended dosing regimen, evaluate their efficacy and safety, and explore their full potential as combination therapy partners [13].
By inhibiting HIF, it is possible to disrupt processes depending on hypoxic conditions, especially in combination with immune checkpoint inhibitors (ICIs) or tyrosine kinase inhibitors (TKIs). This represents a promising therapeutic strategy of ICIs by reversing the immunosuppressive milieu and improving the response to TKIs by limiting VEGF-mediated angiogenesis. Preclinical studies and early-phase clinical trials have started to explore these combinations, showing synergistic effects and supporting further investigation in ccRCC characterized by hypoxia-driven resistance mechanisms.
Currently, a first-line setting (phase 2 LITESPARK-003 study) by using a double treatment with the HIF2α-VEGF inhibitor (belzutifan) in combination with the TKI cabozantinib has shown manageable toxicity [14]. These findings provide the rationale for further randomized trials using belzutifan in combination with TKIs to evaluate improvements in PFS and OS in ccRCC patients.
Phase III clinical trials are currently being conducted to compare the efficacy of belzutifan versus everolimus in patients with advanced RCC previously treated with anti-PD-(L)1 antibodies and VEGF TKIs. These studies demonstrate that belzutifan ameliorates the global health status and quality of life of patients and shows superior efficacy compared to everolimus [15]. Other trials are ongoing: i) LITESPARK-024 to evaluate the efficacy of belzutifan, also in combined therapy with palbociclib [16]; ii) LITESPARK-011, investigating the effects of belzutifan combined with lenvatinib versus belzutifan combined with cabozantinib in patients who did not respond to anti-PD-1/PD-L1 therapy [17]; iii) belzutifan administrated with pembrolizumab in patients who have undergone nephrectomy and/or metastasectomy [18].”
At the end of the chapter, the following sentence was added:
“HIF-mediated pathways that are strongly involved in ccRCC are currently targeted and many studies were conducted not only in models for mRCC but also in clinical trials. As reported above, several trials are now running and findings obtained at the end of these studies will indicate if these strategies will give further improvements for mRCC patients. Last but not least, we will evaluate the side effects that could compromise therapy response, which will require careful management for future clinical applications.”
Chapter 3.2: At the end of the chapter, a new paragraph was added:
“Based on these data, approaches addressed to restore p53 expression should be care-fully evaluated in order to prevent the expression of p53 mutant, which is associated with poor outcomes. The identification of p53 mutations might lead to the generation of alternative therapeutic strategies able to repair p53 lesions or activate the immune system against p53 mutants, killing cancer cells and inducing cancer regression. In ccRCC, the expression of this tumor suppressor is downregulated, therefore the reactivation of wild-type p53 could enhance the therapeutic efficacy in patients refractory to pharmacological treatments. Several clinical trials are being performed or are ongoing to achieve this goal. A phase I study (PYNNACLE) [30] is focused on the PC14586 (rezatapopt) used to correct specifically the mutation Y220C of the TP53 for re-storing its normal conformation [31, 32]. The efficacy of this compound was checked in patients affected by a variety of solid tumors, showing decreased number of circulating tumor cells after treatment. Further investigations are conducted to test the effects of the combined treatment of PC14586 and Pembrolizumab [33]. A joint US-China study is currently underway on the same mutation to evaluate the safety and efficacy of JAB-30355 [34]. Other strategies could be developed, based on the design and construction of TP53-Y220C neo-antigens to enhance the affinity and binding stability to HLA-A0201 molecules to induce increased production of cytotoxic T lymphocytes (CTLs), indicating an improvement in immunogenicity [35, 36].”
Chapter 3.3: At the end of the chapter, a new paragraph was inserted:
“More recently, the combination of the mTORC1/2 inhibitor sapanisertib and the TKI cabozantinib have effectively inhibited tumor growth in patient-derived xenografts (PDXs), some of which were resistant to conventional TKI and immunotherapy combinations. Their action appears to modulate the ERK pathway and downstream transcription factors, leading to cell cycle arrest and apoptosis [42]. Although it appears to be less active when administered alone in refractory mRCC [43] and did not improve the effects of everolimus in advanced or refractory ccRCC [44], sapanisertib has been shown to be effective in the treatment of solid tumors, including ccRCC, in a Phase I clinical study when combined with metformin [45]. Sapanisertib acts by inhibiting the mTOR/AKT/PI3K pathway and shows efficacy in combination with carboplatin and paclitaxel [46] when different pathways are concurrently targeted.
Notably, autophagy is one of the processes regulated by mTOR; in particular, the activation of this kinase inhibits autophagy and promotes growth signals. However, anti-cancer drugs that inhibit mTOR cause the induction of autophagy, which may contribute to cancer progression helping cancer cells to recover energy and remove chemotherapy agents, as widely described below. Therefore, the combined treatment with mTOR and autophagy inhibitors could overcome this issue, limiting the pharmacological resistance.”
Chapter 3.4: At the end of the chapter, a new paragraph was written:
“In this regard, a Phase II open-label multicenter, multicohort study including ccRCC pa-tients was carried out by using the MEK inhibitor cobimetinib in combination with ate-zolizumab. This treatment has shown weak activity in anti-PD-1/PD-L1 treatment-naive renal cell carcinoma and no activity in checkpoint inhibitor-treated patients [53]. These findings indicate that further trials with other MEK inhibitors alone or in combination should be assessed to improve therapeutic efficacy.”
Chapter 3.5: At the end of the chapter, a new paragraph was inserted:
“Currently, no clinical trials are running using Wnt modulators in kidney cancer. Most of data were obtained in “in vitro” and “in vivo” models limiting the translational medicine and the therapeutic potential of these tumor-associated targets.“
Chapter 3.7 was modified as follows (changes are marked in red):
“Taken together, these observations suggest that autophagy in advanced kidney carcinoma is associated with tumor progression and drug resistance by different mechanisms including ERK activation and p53 LOF/GOF. However, before attempting therapies using autophagy inhibitors, it would be appropriate to evaluate whether autophagy is protective or cancer-associated. Preclinical studies have demonstrated that the treatment with autophagy inhibitors alone or in combination with anticancer drugs reduces cancer progression. Based on these observations, several clinical trials using autophagy inhibitors combined with different chemotherapy agents have been conducted, but limited findings were published [80]. Chloroquine and hydroxychloroquine are the only FDA-approved autophagy inhibitors; treatment with these drugs was well tolerated, but further clinical studies are needed to evaluate clinical benefit. Moreover, new more efficient, and specific autophagy inhibitors alone or in combination should be tested in future clinical trials.”
Chapter 3.8 was deeply modified, adding these sentences:
“KTZ not only acts on lncARSR, but also inhibits exosome biogenesis, the formation of intraluminal vesicles as well as vesicle transport through cytoskeletal filaments, by affecting Alix, Neutral sphingomyelinase 2 (nSMase2), and Rab27a [92]. Particular attention is given to proteins involved in secretion processes. Among these, RAB27B is upregulated in RCC cells, especially in sunitinib-resistant ones, whose sensitivity is restored by RAB27B knockout [97]. RNA sequencing and pathway analysis indicated that the function of RAB27B may be mediated by MAPK and VEGF signaling pathways.
Targeting exosome synthesis and trafficking represents a promising novel approach for mRCC therapy. In this context, the combined use of Nexinhib20 and GW4869 with cisplatin or etoposide drugs already employed in first-line chemotherapy for small cell lung cancer could also be applied to RCC. This combination aims to inhibit RAB27A and nSMase2, an enzyme involved in ceramide generation. Additional general inhibitors that could be tested include calpeptin, manumycin A, Y-27632, D-pantethine, and imipramine [98]. Furthermore, exosomes can be exploited as delivery vehicles for interfering RNAs or mimic molecules to counteract the effects of oncogenic ncRNAs or enhance the activity of tumor-suppressive ncRNAs involved in the pathways described below.
Other miRNAs secreted by EVs, such as miR-27a targeting SFRP1, can sustain angiogenesis [99], as it is known that pro-angiogenic miRNAs confer resistance following doxorubicin treatment [100]. TKI- and sorafenib-resistant RCC cells display low exosomal miR-549a levels, leading to increased HIF1α expression, which promotes VEGF secretion, angiogenesis, vascular permeability, and cell migration. This process is further sustained by the activation of the positive feedback VEGFR2–ERK–XPO5 pathway [101].
Although extracellular vesicles (EVs) play a pivotal role in transferring resistance by diffusing their content (proteins, lipids, and nucleic acids) both within the TME and through biological fluids, EV-based therapies are also particularly suitable for drug delivery and therapeutic applications [92]. For example, EVs with low levels of miR-30c-5p, which targets heat-shock protein (HSP) 5, were detected in ccRCC patients, and increased levels of this miRNA affect cancer progression [102]. The influence of exosomes is primarily relevant to cancer-associated fibroblasts (CAFs) in the TME, which are increased in metastatic ccRCC [92]. In this context, CAFs secrete exosomes containing miR-224-5p, which is internalized by ccRCC cells, promoting the migration of metastatic features. Similarly, CAF-derived miR-181d-5p affects RNF43, activating the Wnt/β-catenin signaling pathway. On the other hand, ccRCC cells can release exosomes containing circSAFB2, which inter-feres with the miR-620/JAK1/STAT3 axis promoting polarization of M2 macrophages [92].
Another strategic use of exosomes is applied to immunotherapy and carried out by developing immunotherapeutic vaccines using exosomes derived from RCC cells that induced cytotoxic T lymphocytes against RCC antigens resulting in improved anticancer properties. Other similar approaches were recently used to generate vaccines by using cancer-derived exosomes, which showed increased survival in immunized RCC mice [92].
Although these data support the development of promising future mono or combination therapies aimed at improving overall survival (OS) in patients with advanced ccRCC, other aspects must be considered, particularly the role of EVs in the stimulation of angiogenesis, immune evasion, and tumor progression. Indeed, EVs secreted by RCC cells can inhibit T cell proliferation, natural killer cell activation, and dendritic cell maturation, ultimately leading to reduced sensitivity to ICI therapy [103]. This is partly due to the presence of PD-L1 on the surface of RCC-derived EVs, as well as their ability to induce PD-L1 expression and modulate the immune system [92]. For example, vesicles with miR-224-5p have been shown to promote PD-L1 expression in RCC cells, thereby enhancing resistance to T cell-mediated cytotoxicity [104].
In the light of this data, the targeting of exosome machinery could be a new encouraging strategy to treat metastatic kidney cancer.”
Finally, a summarized paragraph at the end of Results and Discussion was inserted:
“Overall, data here reported indicate that cancer cells adopt many strategies to evade cell death leading to drug insensitivity and cancer progression. Despite great progress obtained, there is still much work to do in order to set up specific therapies capable of eluding pharmacological resistance. Most promised studies were mainly carried out in “in vitro” and in pre-clinical animal models limiting therapeutic availability in a short time. Future clinical studies should be performed to test the efficacy of therapeutic targets that gave positive responses in pre-clinical investigations.”
- In line 497 authors introduce Nrf2 which is a known key factor for therapy resistance in renal cell carcinoma (PMID: 39769005). This topic requires a larger discussion.
Answer: We thank the Reviewer for this valuable suggestion. We have addressed the crucial role of NRF2 in the context of ferroptosis by adding the following text to the revised manuscript.
“In ccRCC, the constitutive activity of NRF2 and the following constitutive overexpression of the genes under its transcriptional control is triggered by ROS, determining the switch from a protective to tumorigenic profile in an attempt to limit oxidative stress and inflammation [110]. This condition is exacerbated by drugs such as cisplatin, increasing the activation of the NRF2/ARE pathway and mutations in the genes KEAP1 and CUL3, which in normal conditions led to NRF2 inactivation by ubiquitin [110]. Several genes are regulated by this transcription factor and engaged in the detoxification processes, many of them are involved also in ferroptosis. In addition to SLC7A11, we found glutathione S-transferases and heme oxygenase- (HO-1)1, to mention just a few.
NRF2 stabilization depends on the expression of dipeptidyl peptidase 9 (DPP9), which is upregulated in ccRCC and correlates with advanced tumor stage and poor prognosis. DPP9 overexpression inhibits ferroptosis and induces sorafenib resistance in ccRCC cells, in a mechanism involving the activation of the NRF2/SLC7A11 axis [111].
We underline that NRF2 represents a crucial node interconnected with the deregulated PI3K/Akt pathway, which modulates the expression of HO-1 and autophagy, regulating the expression of SQSTM1 [110]; on the other hand, mutations in the NRF2 promoter upregulating it were frequent in patients with metastatic ccRCC and not responding to VEGF-targeted therapy [111].
It is reasonable to assume that promoting the inhibition of this factor induces anti-cancer effects. Indeed, it has been shown that the miR-200a-3p/141-3p/KEAP1 axis pro-motes the proteasomal degradation of NFE2 [110]. Decreased levels of miR-200 were observed in RCC patients with inactive Fumarase, known to be associated with accumulation of fumarate affecting KEAP1 and NRF2 increase.
Since NRF2 inhibitors have proven ineffective, modulating it by targeting its regulatory pathways could be considered. Among the possible candidates, we mention dimethyl fumarate and KEAP1 knockdown/silencing, which may ameliorate tumor sensitivity in axitinib- or sunitinib-resistant patients or in association with the VEGFR inhibitor pazopanib. Interestingly, anticancer properties demonstrated by the ginsenoside Rh4, increasing ferroptosis in RCC, appear mediated by mechanisms involving mainly NRF2 [113].”
- Figure must be improved since are excessively blurry.
Answer: We apologize for the issue. In the original file, the figures do not appear blurred; however, we have made an effort to further improve their quality. The original version of the figures has also been attached separately to the manuscript in TIFF format at 600 dpi. Please let us know if there are still any issues.
- There are strange symbols in the text.
Answer: We have extensively corrected the format and the errors.
- The conclusion section should be totally rewritten since it actually does not add anything as it is now. Please clearly focus on actual state-of-art and future perspectives.
Answer: as requested the Conclusion was rewritten as follows:
“In recent years, understanding the molecular mechanisms behind resistance has opened new avenues for addressing the issue. Pharmacological insensitivity of cancer cells leads to the failure of current therapies with limited overall survival for mRCC patients. Genetic mutations activating oncogenic signals or inhibiting tumor suppressor factors are the main causes of resistance after prolonged treatment. Other players such as cell adaptation to changes in TME, alteration of cellular transport as well as the activation or the shut-down of different biological processes including EMT, autophagy, and ferroptosis are also important. Patients who have failed first-line therapy should benefit from targeted more efficient second-line treatments to improve their survival and quality of life. Therefore, new investigations should be carried out to understand resistance mechanisms to detect new therapeutic targets that could overcome this problem. NEXT-generation drugs targeting patient-specific cancer factors and multitarget therapies could prevent drug resistance. New strategies addressed to the activation of immunity response against tumor cells, also could enhance the benefits of anticancer treatments. If these approaches are effective, they could open a new era for the treatment of metastatic renal carcinoma and other cancers re-sistant to conventional therapies.”
Comments on the Quality of English Language
The manuscript requires a native English-speaker revision.
Answer: done.

Reviewer 2 Report
Comments and Suggestions for Authors
- The review “Molecular Mechanisms of Drug Resistance in Kidney Carcinoma ” is comprehensive and well-organized. Still, it would benefit from a clearer graphical summary of all discussed pathways at the end of the manuscript.
- Some sections are highly detailed, and including so many mechanisms may overwhelm readers. A table summarizing each resistance mechanism, drug, and potential target would help.
3.The discussion on HIF2 inhibitors is well-articulated, but the potential clinical relevance of combining HIF2 inhibitors with ICIs or TKIs deserves a stronger emphasis.
- In the p53 section, the manuscript might benefit from mentioning emerging pharmacological p53 reactivators under development or trials.
- The section on mTOR resistance is robust, yet referencing dual mTORC1/2 inhibitors (e.g., sapanisertib) could be a valuable addition.
- While the dual role is acknowledged in the autophagy section, the manuscript could better explain how to balance pro-survival and pro-death autophagy modulation in therapy.
- The role of exosomes in transferring resistance is fascinating. However, more discussion is needed on the translational potential of exosome inhibition.
- The ncRNA section is thorough but dense. Using figures to depict key lncRNA-miRNA-mRNA regulatory loops could enhance clarity.
- There are several formatting inconsistencies—such as spacing issues (e.g., “HIF2 ” instead of “HIF2α”) and missing Greek letters or subscripts.
- Include latest citations relevant to your work.
Author Response
REPLY to the Reviewer 2
- The review “Molecular Mechanisms of Drug Resistance in Kidney Carcinoma” is comprehensive and well-organized. Still, it would benefit from a clearer graphical summary of all discussed pathways at the end of the manuscript.
Answer: We have revised the abstract, as the number of involved pathways is extensive. For this reason, we opted for a new formulation that we believe offers a clearer and more concise overview. Please let us know if this version is more suitable.
- Some sections are highly detailed, and including so many mechanisms may overwhelm readers. A table summarizing each resistance mechanism, drug, and potential target would help.
Answer: As suggested by the Reviewer a new table summarizing drug resistance mechanisms was inserted in the text.
- The discussion on HIF2 inhibitors is well-articulated, but the potential clinical relevance of combining HIF2 inhibitors with ICIs or TKIs deserves a stronger emphasis.
Answer: The following phrases has been added to better highlight the relevance of clinical trials:
“Some HIF2α inhibitors, such as DFF332, require further investigation to establish a recommended dosing regimen, evaluate their efficacy and safety, and explore their full potential as combination therapy partners [13].
By inhibiting HIF, it is possible to disrupt processes depending on hypoxic conditions, especially in combination with immune checkpoint inhibitors (ICIs) or tyrosine ki-nase inhibitors (TKIs). This represents a promising therapeutic strategy of ICIs by reversing the immunosuppressive milieu and improving the response to TKIs by limiting VEGF-mediated angiogenesis. Preclinical studies and early-phase clinical trials have started to explore these combinations, showing synergistic effects and supporting further investigation in ccRCC characterized by hypoxia-driven resistance mechanisms.
Currently, a first-line setting (phase 2 LITESPARK-003 study) by using a double treatment with the HIF2α-VEGF inhibitor (belzutifan) in combination with the TKI cabozantinib has shown manageable toxicity [14]. These findings provide the rationale for further randomized trials using belzutifan in combination with TKIs to evaluate im-provements in PFS and OS in ccRCC patients.
Phase III clinical trials are currently being conducted to compare the efficacy of belzu-tifan versus everolimus in patients with advanced RCC previously treated with anti-PD-(L)1 antibodies and VEGF TKIs. These studies demonstrate that belzutifan ameliorates the global health status and quality of life of patients and shows superior efficacy compared to everolimus [15]. Other trials are ongoing: i) LITESPARK-024 to evaluate the efficacy of belzutifan, also in combined therapy with palbociclib [16]; ii) LITESPARK-011, investigating the effects of belzutifan combined with lenvatinib versus belzutifan combined with cabozantinib in patients who did not respond to anti-PD-1/PD-L1 therapy [17]; iii) belzutifan administrated with pembrolizumab in patients who have undergone nephrectomy and/or metastasectomy [18].”
- In the p53 section, the manuscript might benefit from mentioning emerging pharmacological p53 reactivators under development or trials.
Answer: As suggested, we included several studies on the topic:
“Based on these data, approaches addressed to restore p53 expression should be care-fully evaluated in order to prevent the expression of p53 mutant, which is associated with poor outcomes. The identification of p53 mutations might lead to the generation of alternative therapeutic strategies able to repair p53 lesions or activate the immune system against p53 mutants, killing cancer cells and inducing cancer regression. In ccRCC, the expression of this tumor suppressor is downregulated, therefore the reactivation of wild-type p53 could enhance the therapeutic efficacy in patients refractory to pharmacological treatments. Several clinical trials are being performed or are ongoing to achieve this goal. A phase I study (PYNNACLE) [30] is focused on the PC14586 (rezatapopt) used to correct specifically the mutation Y220C of the TP53 for re-storing its normal conformation [31, 32]. The efficacy of this compound was checked in patients affected by a variety of solid tumors, showing decreased number of circulating tumor cells after treatment. Further investigations are conducted to test the effects of the combined treatment of PC14586 and Pembrolizumab [33]. A joint US-China study is currently underway on the same mutation to evaluate the safety and efficacy of JAB-30355 [34]. Other strategies could be developed, based on the design and construction of TP53-Y220C neo-antigens to enhance the affinity and binding stability to HLA-A0201 molecules to induce increased production of cytotoxic T lymphocytes (CTLs), indicating an improvement in immunogenicity [35, 36].”
- The section on mTOR resistance is robust, yet referencing dual mTORC1/2 inhibitors (e.g., sapanisertib) could be a valuable addition.
Answer: Thank you for your valuable suggestion. Considering your feedback, we have included the following paragraph in the revised manuscript:
“More recently, the combination of the mTORC1/2 inhibitor sapanisertib and the TKI cabozantinib have effectively inhibited tumor growth in patient-derived xenografts (PDXs), some of which were resistant to conventional TKI and immunotherapy combinations. Their action appears to modulate the ERK pathway and downstream transcription factors, leading to cell cycle arrest and apoptosis [42]. Although it appears to be less active when administered alone in refractory mRCC [43] and did not improve the effects of everolimus in advanced or refractory ccRCC [44], sapanisertib has been shown to be effective in the treatment of solid tumors, including ccRCC, in a Phase I clinical study when combined with metformin [45]. Sapanisertib acts by inhibiting the mTOR/AKT/PI3K pathway and shows efficacy in combination with carboplatin and paclitaxel [46] when different pathways are concurrently targeted.
Notably, autophagy is one of the processes regulated by mTOR; in particular, the activation of this kinase inhibits autophagy and promotes growth signals. However, anti-cancer drugs that inhibit mTOR cause the induction of autophagy, which may contribute to cancer progression helping cancer cells to recover energy and remove chemotherapy agents, as widely described below. Therefore, the combined treatment with mTOR and autophagy inhibitors could overcome this issue, limiting the pharmacological resistance.”
- While the dual role is acknowledged in the autophagy section, the manuscript could better explain how to balance pro-survival and pro-death autophagy modulation in therapy.
Answer: We appreciate the reviewer’s comment and agree that clarifying this point is important. We have modified the title: “Turning on and off” autophagy and included the following paragraph:
“Autophagy plays a dual role in cancer, acting either as a survival mechanism that pro-motes tumor cell resistance under stress conditions or as a cell death pathway when excessively activated. In the context of RCC, therapeutic strategies should aim to modulate autophagy in a context-dependent manner. For instance, in cases where autophagy sup-ports tumor cell survival and drug resistance, combining standard therapies with autophagy inhibitors may enhance treatment efficacy. Also, anticancer effects can be potentiated by coupling PI3K/mTOR inhibitors as NVP-BEZ23 and autophagy inhibitors [70]. Conversely, when autophagy leads to cell death, therapeutic approaches that promote autophagic flux could be beneficial [71]. Therefore, careful evaluation of autophagy status in RCC cells, possibly through biomarkers or functional assays, is crucial to determine whether to inhibit or stimulate the autophagic process as part of a personalized treatment strategy. Indeed, it has been reported that the expression levels of apoptosis-related genes did not show any significant association with the clinicopathological parameters of ccRCC. In contrast, elevated mRNA expression of autophagy-related genes (i.g. ATG4, GABARAP, and p62) was linked to earlier tumor stages, smaller tumor dimensions, and specifically for ATG4 and p62 improved disease-specific survival over five years [72]. Consistently, reduced protein levels of p62, indicative of enhanced autophagic activity, correlated with less advanced disease, fewer metastatic events, and better long-term prognosis, suggesting that in ccRCC the activation of the autophagic machinery at the transcriptional level is associated with increased autophagic flux, occurring independently of the AMPK/mTOR pathway. Notably, ccRCC often exhibits either a monoallelic deletion or mutation of the autophagy-related gene ATG7, and diminished expression of autophagy-associated genes has been linked to disease progression. In line with this, ccRCC tumor tissues show decreased protein levels of both ATG7 and Beclin 1, supporting the notion that in some cases autophagy functions as a tumor-suppressive mechanism during ccRCC development. Moreover, the data highlight a potential role for constitutive autophagic degradation of HIF2α by an unrecognized pathway contributing to tumor suppression [73].”
The balance between protective or cancer associated autophagy should be carefully evaluated before treating patients with autophagy inhibitor; therefore, at the end of the chapter 3.7, we have modified the last paragraph as follows:
“Taken together, these observations suggest that autophagy in advanced kidney carcinoma is associated with tumor progression and drug resistance by different mechanisms including ERK activation and p53 LOF/GOF. However, before attempting therapies using autophagy inhibitors, it would be appropriate to evaluate whether autophagy is protective or cancer-associated. Preclinical studies have demonstrated that the treatment with autophagy inhibitors alone or in combination with anticancer drugs reduces cancer progression. Based on these observations, several clinical trials using autophagy inhibitors combined with different chemotherapy agents have been conducted, but limited findings were published [80]. Chloroquine and hydroxychloroquine are the only FDA-approved autophagy inhibitors; treatment with these drugs was well tolerated, but further clinical studies are needed to evaluate clinical benefit. Moreover, new more efficient, and specific autophagy inhibitors alone or in combination should be tested in future clinical trials.”
- The role of exosomes in transferring resistance is fascinating. However, more discussion is needed on the translational potential of exosome inhibition.
Answer: An in-depth discussion of the relevance of exosomes in RCC is provided below, including the following points (include in the paragraph “Drug removal by transporters and exosome machinery”):
“Exosomes are commonly used as drug delivery vehicles in cancer therapy; however, they also have a dark side, contributing to tumor development by influencing stromal and immune cells in the TME, and promoting cancer progression, metastasis, and therapy resistance, particularly in kidney cancer [92]. Exosome trafficking is an important mechanism of cellular communication that may enable chemotherapy-resistant cells to induce pharmacological resistance in drug-sensitive cells, thereby rendering treatment ineffective.”
“KTZ not only acts on lncARSR, but also inhibits exosome biogenesis, the formation of intraluminal vesicles as well as vesicle transport through cytoskeletal filaments, by affecting Alix, Neutral sphingomyelinase 2 (nSMase2), and Rab27a [92]. Particular attention is given to proteins involved in secretion processes. Among these, RAB27B is upregulated in RCC cells, especially in sunitinib-resistant ones, whose sensitivity is restored by RAB27B knockout [97]. RNA sequencing and pathway analysis indicated that the function of RAB27B may be mediated by MAPK and VEGF signaling pathways.
Targeting exosome synthesis and trafficking represents a promising novel approach for mRCC therapy. In this context, the combined use of Nexinhib20 and GW4869 with cisplatin or etoposide drugs already employed in first-line chemotherapy for small cell lung cancer could also be applied to RCC. This combination aims to inhibit RAB27A and nSMase2, an enzyme involved in ceramide generation. Additional general inhibitors that could be tested include calpeptin, manumycin A, Y-27632, D-pantethine, and imipramine [98]. Furthermore, exosomes can be exploited as delivery vehicles for interfering RNAs or mimic molecules to counteract the effects of oncogenic ncRNAs or enhance the activity of tumor-suppressive ncRNAs involved in the pathways described below.
Other miRNAs secreted by EVs, such as miR-27a targeting SFRP1, can sustain angiogenesis [99], as it is known that pro-angiogenic miRNAs confer resistance following doxorubicin treatment [100]. TKI- and sorafenib-resistant RCC cells display low exosomal miR-549a levels, leading to increased HIF1α expression, which promotes VEGF secretion, angiogenesis, vascular permeability, and cell migration. This process is further sustained by the activation of the positive feedback VEGFR2–ERK–XPO5 pathway [101].
Although extracellular vesicles (EVs) play a pivotal role in transferring resistance by diffusing their content (proteins, lipids, and nucleic acids) both within the TME and through biological fluids, EV-based therapies are also particularly suitable for drug delivery and therapeutic applications [92]. For example, EVs with low levels of miR-30c-5p, which targets heat-shock protein (HSP) 5, were detected in ccRCC patients, and increased levels of this miRNA affect cancer progression [102]. The influence of exosomes is primarily relevant to cancer-associated fibroblasts (CAFs) in the TME, which are increased in metastatic ccRCC [92]. In this context, CAFs secrete exosomes containing miR-224-5p, which is internalized by ccRCC cells, promoting the migration of metastatic features. Similarly, CAF-derived miR-181d-5p affects RNF43, activating the Wnt/β-catenin signaling pathway. On the other hand, ccRCC cells can release exosomes containing circSAFB2, which inter-feres with the miR-620/JAK1/STAT3 axis promoting polarization of M2 macrophages [92].
Another strategic use of exosomes is applied to immunotherapy and carried out by developing immunotherapeutic vaccines using exosomes derived from RCC cells that induced cytotoxic T lymphocytes against RCC antigens resulting in improved anticancer properties. Other similar approaches were recently used to generate vaccines by using cancer-derived exosomes, which showed increased survival in immunized RCC mice [92].
Although these data support the development of promising future mono or combination therapies aimed at improving overall survival (OS) in patients with advanced ccRCC, other aspects must be considered, particularly the role of EVs in the stimulation of angiogenesis, immune evasion, and tumor progression. Indeed, EVs secreted by RCC cells can inhibit T cell proliferation, natural killer cell activation, and dendritic cell maturation, ultimately leading to reduced sensitivity to ICI therapy [103]. This is partly due to the presence of PD-L1 on the surface of RCC-derived EVs, as well as their ability to induce PD-L1 expression and modulate the immune system [92]. For example, vesicles with miR-224-5p have been shown to promote PD-L1 expression in RCC cells, thereby enhancing resistance to T cell-mediated cytotoxicity [104].
In the light of this data, the targeting of exosome machinery could be a new encouraging strategy to treat metastatic kidney cancer”
- The ncRNA section is thorough but dense. Using figures to depict key lncRNA-miRNA-mRNA regulatory loops could enhance clarity.
Answer: We have included the new Figure 3 in the paragraph “Mechanisms engaging lncRNAs and circRNAs in the chemoresistance”.
- There are several formatting inconsistencies such as spacing issues (e.g., “HIF2” instead of “HIF2α”) and missing Greek letters or subscripts.
Answer: We have extensively corrected the format and the errors.
- Include latest citations relevant to your work.
Answer: We have significantly expanded the reference list, as the reviewer can see. If any additional references deemed relevant are still missing, please do not hesitate to point them out.

Round 2
Reviewer 1 Report
Comments and Suggestions for Authors
The authors correctly addressed all the raised concerns, thus the manuscript can be accepted for publication.